# Conserved noncoding transcription and core promoter regulatory code in early *Drosophila* development

**Philippe J Batut[†], Thomas R Gingeras[‡]\***

Watson School of Biological Sciences, Cold Spring Harbor Laboratory, New York, United States

**Abstract** Multicellular development is driven by regulatory programs that orchestrate the transcription of protein-coding and noncoding genes. To decipher this genomic regulatory code, and to investigate the developmental relevance of noncoding transcription, we compared genome-wide promoter activity throughout embryogenesis in 5 *Drosophila* species. Core promoters, generally not thought to play a significant regulatory role, in fact impart restrictions on the developmental timing of gene expression on a global scale. We propose a hierarchical regulatory model in which core promoters define broad windows of opportunity for expression, by defining a range of transcription factors from which they can receive regulatory inputs. This two-tiered mechanism globally orchestrates developmental gene expression, including extremely widespread noncoding transcription. The sequence and expression specificity of noncoding RNA promoters are evolutionarily conserved, implying biological relevance. Overall, this work introduces a hierarchical model for developmental gene regulation, and reveals a major role for noncoding transcription in animal development.

DOI: https://doi.org/10.7554/eLife.29005.001

**\*For correspondence:**
gingeras@cshl.edu

**Present address:** [†]Lewis-Sigler Institute, Princeton University, Princeton, United States; [‡]Functional Genomics, Cold Spring Harbor Laboratory, Cold Spring Harbor, New York, United States

## Introduction

Development in metazoans is orchestrated by complex gene regulatory programs encoded in the sequence of the genome (*Lewis, 1978*; *Nüsslein-Volhard and Wieschaus, 1980*; *Levine and Davidson, 2005*; *Peter and Davidson, 2011*). The expression of thousands of protein-coding and noncoding genes, in precise spatiotemporal patterns, progressively refines the organization of embryonic structures and specifies the differentiation of specialized cell types. In *Drosophila*, many of the master genes governing early development (*Lewis, 1978*; *Nüsslein-Volhard and Wieschaus, 1980*) encode regulators of transcription (*St Johnston and Nüsslein-Volhard, 1992*; *Levine and Hoey, 1988*; *Hoey and Levine, 1988*; *Desplan et al., 1988*), and transcriptional regulation largely accounts for embryo patterning (*St Johnston and Nüsslein-Volhard, 1992*; *Segal et al., 2008*).

The regulatory code that specifies these programs, however, remains poorly understood. Sequences that bind transcriptional activators and repressors, known as enhancers (*Banerji et al., 1981*; *Banerji et al., 1983*; *Zinn et al., 1983*; *Small et al., 1991*; *Spitz and Furlong, 2012*; *Schaffner, 2015*; *Benoist and Chambon, 1980*), are generally thought to be the primary determinants of gene expression specificity. Sequences that serve as docking sites for the basal transcription machinery, the core promoters (*Lenhard et al., 2012*), are usually assumed to be structural elements that contribute little or no regulatory information.

Indeed, core promoters contain sequence motifs (e.g., TATA boxes) that serve as a platform for RNA Pol II initiation at transcription start sites (TSSs), but are not by themselves sufficient to induce transcription (*Spitz and Furlong, 2012*; *Lenhard et al., 2012*). Sequence-specific activators and repressors, collectively designated as transcription factors (TFs), bind to enhancers and foster the

assembly of the basal transcription machinery at associated core promoters (*Spitz and Furlong, 2012*; *Lenhard et al., 2012*). Beyond these general principles, the syntax of the code, and in particular the functional relationship between these two classes of elements, remains obscure. Interacting core promoter-enhancer pairs may be directly adjacent (*Zinn et al., 1983*; *Benoist and Chambon, 1980*; *Zabidi et al., 2015*), or may be located hundreds of kilobases apart in metazoan genomes (*Spitz and Furlong, 2012*; *Lettice et al., 2003*), and the rules enforcing specific interactions are unknown (*Spitz and Furlong, 2012*; *Lenhard et al., 2012*). There is evidence that core promoters can influence the expression specificity of some genes (*Zabidi et al., 2015*; *Goodrich and Tjian, 2010*; *Haberle et al., 2014*), but so far this has not been systematically studied in the context of development. Understanding the basis of transcriptional control requires parsing out these complex interactions.

In addition to delineating the rules of gene regulation, it is necessary to expand the concept of gene (*Gerstein et al., 2007*; *Gingeras, 2007*) to include all the noncoding loci that may control developmental processes. Indeed, it has become increasingly clear that noncoding transcription is very prevalent in Eukaryotes (*Kapranov et al., 2007*; *Djebali et al., 2012*; *Graveley et al., 2011*; *Gerstein et al., 2010*; *Young et al., 2012*; *Derrien et al., 2012*), and both genetic and biochemical studies have unambiguously established long noncoding RNAs (lncRNAs) as functional components of the cellular machinery (*Augui et al., 2011*; *Ulitsky and Bartel, 2013*; *Guttman and Rinn, 2012*; *Ponting et al., 2009*). Exhaustively annotating noncoding transcripts, and identifying those with biologically relevant functions, is crucial to our understanding of development.

Deciphering the meaning of regulatory sequences, or assessing the biological relevance of lncRNA genes, are daunting tasks that require innovative strategies. The use of genome-wide functional assays in a phylogenetic framework is a powerful and general approach to such questions (*Tsankov et al., 2011*; *Schmidt et al., 2010*; *Stefflova et al., 2013*; *He et al., 2011*). Indeed, direct measurements of complex genome functions in multiple species provide a sort of genomic Rosetta Stone from which the underlying code can begin to be parsed out.

Here, we used high-throughput TSS mapping in tightly resolved time series to establish genome-wide promoter activity profiles throughout embryonic development in 5 *Drosophila* species spanning 25–50 million years (MY) of evolution. Combining TSS identification at single-nucleotide resolution (*Batut et al., 2013*) with quantitative measurements of developmental expression patterns, we uncovered unique features of expression timing and core promoter structure to generate novel insights into transcriptional regulation.

We report that distinct types of core promoters are selectively active in three broad phases of embryonic development: specific combinations of core motifs mediate transcription during Early, Intermediate and Late embryogenesis. Each individual class of core promoters is functionally associated with distinct sets of transcription factors. We propose a two-tier model of transcriptional control in which core promoters and enhancers mediate, respectively, coarse-grained and fine-grained developmental regulation.

We also show that noncoding transcription is far more widespread than anticipated in *Drosophila*, with 3973 promoters driving the expression of lncRNAs during embryogenesis. The analysis of these core promoters, most of which are currently unannotated, shows that they largely share the structural and functional properties of their counterparts at protein-coding genes. Through the analysis of their fine structure and sequence conservation, we demonstrate that evolutionarily conserved lncRNAs promoters are under strong purifying selection at the levels of primary sequence and expression specificity. We functionally characterize the *FBgn0264479* locus, which expresses a lncRNA in a highly conserved spatiotemporal pattern suggestive of a role in early dorsoventral patterning.

In summary, these results uncover a major active role for core promoters in regulating transcription, by defining windows of opportunity for activation by enhancer sequences. They also reveal a vastly underappreciated aspect of developmental transcriptomes, by showing that noncoding transcription is extremely prevalent, tightly regulated and, crucially, deeply conserved.

## Results

### Global multispecies profiling of developmental promoter activity

To explore the genome-wide dynamics of transcriptional regulation and their evolution, we generated developmental transcriptome profiles at both very high temporal resolution (1 hr) and high sequence coverage (137–180 million read pairs per species) for 5 *Drosophila* species spanning 25–50 million years of evolution: *D. melanogaster*, *D. simulans*, *D. erecta*, *D. ananassae* and *D. pseudoobscura* (*Figure 1A*; total of 120 samples). We focused on embryonic development, a crucial period during which the body plan is established and all larval organs are generated. This data allows direct, genome-wide comparisons of promoter activity in a phylogenetic framework (*Figure 1B*).

In order to map transcription start sites (TSSs) with single-base resolution and accurately measure the activity of individual promoters, we used a high-fidelity method called RAMPAGE (*Batut et al., 2013*) based on high-throughput sequencing of 5′-complete complementary DNA (cDNA) molecules. It offers unprecedented specificity and detection sensitivity for TSSs, and its multiplexing capabilities allow for the seamless acquisition of high-resolution developmental time series (*Batut et al., 2013*). Since eukaryotic promoters often allow productive transcription initiation from multiple positions (*Batut et al., 2013*; *Carninci et al., 2006*; *Valen et al., 2009*; *Hoskins et al., 2011*), we use a dedicated peak-calling algorithm to group neighboring TSSs into TSS clusters (TSCs) corresponding to individual promoters (*Batut et al., 2013*; *Batut and Gingeras, 2013*).

For each species, we identified $2.2 \times 10^4$ to $2.7 \times 10^4$ high-confidence TSCs. The narrow distribution of raw RAMPAGE signal (*Figure 1C* and *Figure 1—figure supplement 1*) and of TSCs (*Figure 1—figure supplement 2*) over annotated loci confirms our very high specificity for true TSSs. The quantification of promoter expression levels is highly reproducible across biological replicates (*Figure 1D–E*). Importantly, paired-end sequencing of cDNAs allows for evidence-based assignment of novel TSCs to existing gene annotations, and provides valuable information about overall transcript structure (*Figure 1B*). We can thus attribute 82% of *D. melanogaster* TSCs to annotated genes, the remaining 18% potentially driving the expression of unannotated non-protein-coding transcripts.

The comparison of biological replicates for the *D. melanogaster* time series confirms our ability to quantitatively measure promoter expression dynamics throughout development. Indeed, this analysis shows excellent reproducibility (Pearson $R^2 = 0.95$) for TSCs with maximum expression $\geq$25 reads per million (RPM, *Figure 2A* and *Figure 2—figure supplement 1*), and very good reproducibility (Pearson $R^2 = 0.92$) with maximum expression $\geq$10 RPM (*Figure 2—figure supplement 1*).

Post-synchronization of developmental series was achieved by global alignments of all time series to the *D. melanogaster* reference to maximize the overall similarity between genome-wide expression profiles (*Goltsev and Papatsenko, 2009*; *Kalinka et al., 2010*) (*Figure 2—figure supplement 2*). The resulting alignments for well-known developmental genes, used here as diagnostic markers, confirm the very high quality of the global alignments (*Figure 2B* and *Figure 2—figure supplement 3*). For all genes with one-to-one orthologs, the expression profiles are overall tightly conserved across species (*Figure 2C*), but with substantial gene-to-gene variability (*Figure 2—figure supplement 4*): *hunchback*, for instance, displays considerable conservation in the expression of both of its promoters (Pearson $R^2$ of 0.88 and 0.97), whereas the *RpL19* promoter shows rapid divergence ($R^2 = 0.08$) (*Figure 2—figure supplement 4*).

We found a strong relationship between selective constraints on expression specificity and gene function: indeed, the degree of expression divergence differs substantially between Gene Ontology (GO) annotation categories (*Figure 2D*). Functions related to the regulation of transcription and splicing display the strongest conservation of expression, in accordance with the known molecular function of many master regulators of early development. Categories related to the core translational machinery and cytoskeletal structures display much more plastic expression specificities.

The high similarity of biological replicates, the accuracy of inter-species alignments for well-known developmental genes, and the biological features of evolutionary divergence patterns, together confirm our ability to accurately quantify promoter expression and its variation across species. Our observations also highlight the role of systems-level selective constraints, such as those acting on particular functional gene classes (*Figure 2D*), in shaping the evolution of gene expression.

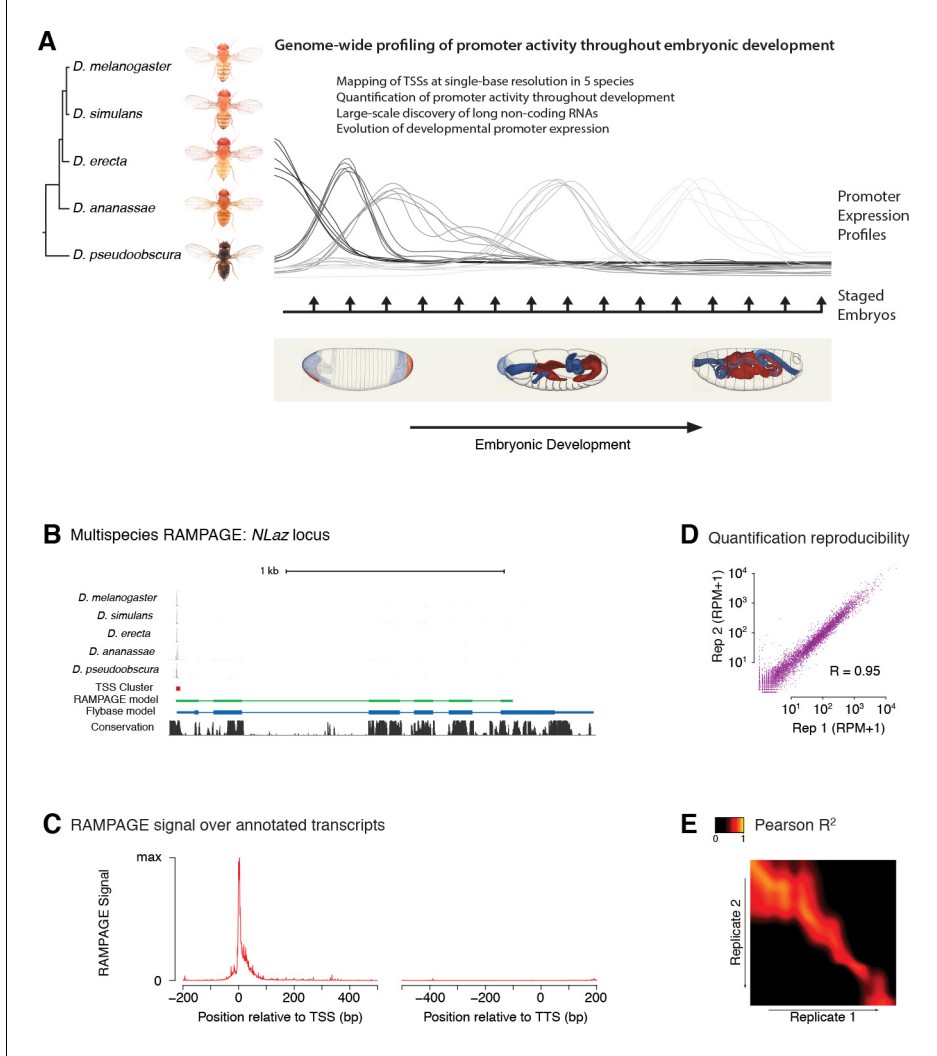

**Figure 1.** Comparative profiling of embryonic promoter activity. (**A**) Genome-wide TSS usage maps were generated by RAMPAGE assays for developmental series in five species. These datasets constitute single-nucleotide measurements of transcriptional activity at 1 hr resolution throughout embryonic development, and a global survey of its evolution over 25-50MY. Fly photographs (N. Gompel) and embryo drawings (V. Hartenstein, CSHL Press 1993) reproduced with permission from FlyBase. (**B**) This panel illustrates the 5-species RAMPAGE data at a conserved locus (*NLaz*). The top five tracks show the density of RAMPAGE read 5' ends over the locus. The non-*melanogaster* data was aligned to the appropriate genome references and projected for visualization onto the orthologous positions in *D. melanogaster*. The next tracks show the Transcription Start Site Cluster (TSC) called from this 5' end data in *D. melanogaster*, and the partial transcript model reconstructed for this TSC from the full sequencing reads. Note the agreement with the transcript annotated in FlyBase, in blue, for the portion covered by the data. The last track shows phastCons sequence conservation scores. (**C**) Metagene profile of RAMPAGE read 5' ends over FlyBase-annotated mature transcripts (introns excluded). (**D**) Reproducibility of RAMPAGE signal quantification for individual TSCs (n = 9,299) for two biological replicates of the first *D. melanogaster* time point (0–1 hr). TSCs with no signal in either replicate were excluded. (**E**) Correlation matrix for the *D. melanogaster* time series biological replicates. The heatmap shows the correlation of TSC expression (n = 24,832 TSCs) after upsampling, smoothing and alignment of the time series (see Materials and methods).

DOI: https://doi.org/10.7554/eLife.29005.002

The following figure supplements are available for figure 1:

**Figure supplement 1.** Distribution or raw RAMPAGE signal over transcript annotations.
DOI: https://doi.org/10.7554/eLife.29005.003
**Figure supplement 2.** Distribution or RAMPAGE peaks over transcript annotations.
DOI: https://doi.org/10.7554/eLife.29005.004

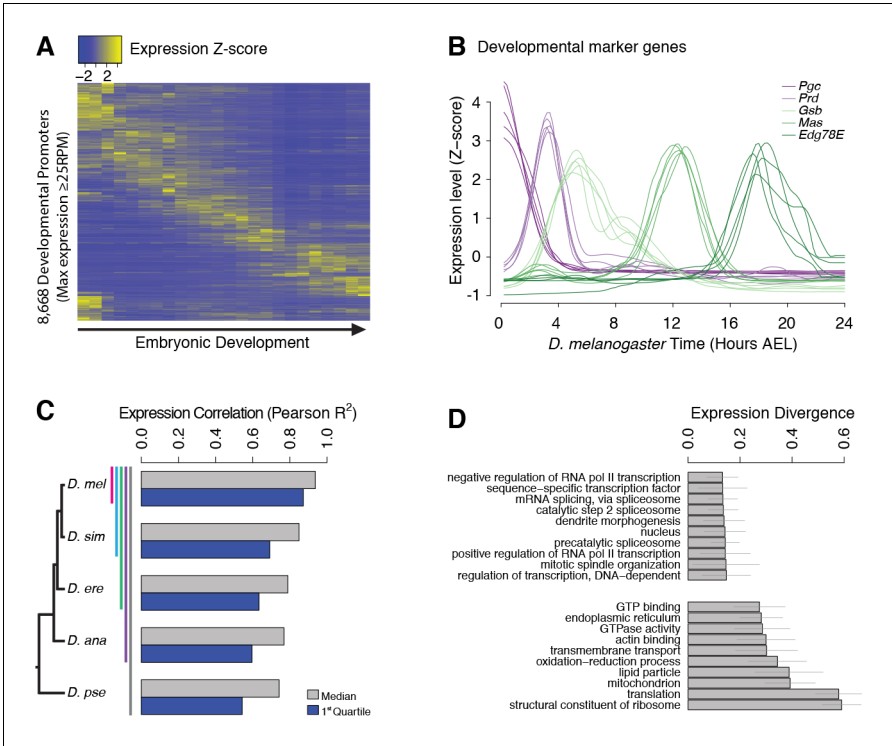

**Figure 2.** Evolutionary divergence of the developmental timing of promoter activity. (**A**) Hierarchically clustered expression profiles for individual *D. melanogaster* promoters throughout embryogenesis (24 time points; replicate 1). Only promoters with a maximum expression level ≥25 RPM (n = 8,668), for which quantification reproducibility is very high, are included. (**B**) Expression profiles for five developmental marker genes, after global alignment of all time series to *D. melanogaster* (see Materials and methods). The five curves for each gene correspond to the five species. (**C**) Conservation of the temporal expression profiles of individual promoters for all subclades of the phylogeny (subclades include all species descended from a common ancestor, and are coded by color bars next to the tree). For each subclade, we computed the average correlation coefficient between all pairs of species for each individual gene. The graph shows the median and first quartile over all genes with orthologs throughout the subclade. (**D**) The evolutionary divergence of expression specificity varies widely between Gene Ontology (GO) categories ($p<10^{-16}$, one-way ANOVA). For each gene with a GO annotation, orthologs in all five species, maximum expression ≥25 RPM and expression changes ≥5 fold (n = 2,690), we computed a measure of overall divergence across the clade (see Materials and methods). The bar plot shows the average divergence by GO category, for the 20 categories with the lowest (top) and greatest (bottom) divergence. Error bars show ±1 standard deviation.

DOI: https://doi.org/10.7554/eLife.29005.005

The following figure supplements are available for figure 2:

**Figure supplement 1.** Reproducibility of expression time series.

DOI: https://doi.org/10.7554/eLife.29005.006

**Figure supplement 2.** Time series alignment by time-warping of gene expression profiles.

DOI: https://doi.org/10.7554/eLife.29005.007

**Figure supplement 3.** Global alignment of time series to *D. melanogaster* reference.

DOI: https://doi.org/10.7554/eLife.29005.008

**Figure supplement 4.** Evolutionary conservation of TSC expression profiles.

DOI: https://doi.org/10.7554/eLife.29005.009

## Core promoter structure defines broad developmental phases of gene expression

We leveraged our multispecies expression data to study promoter structure-function relationships, and thus gain insights into the regulatory code that determines developmental gene expression. We focused on a set of 3462 promoters functionally active in all species that we classified either as housekeeping (<5 fold variation throughout development) or as developmentally regulated (≥60%

of total expression within an 8 hr window). The developmentally regulated group was further clustered based on temporal correlation, yielding a total of 9 distinct expression clusters (*Figure 3A*, see Materials and methods). These thresholds were chosen to maximize the total number of promoters included and the similarity of profiles within each cluster, while still yielding clusters large enough for statistical analysis.

These nine expression clusters fall into three main groups, defined by the core motif composition of the promoters (*Figure 3B*). Indeed, we unexpectedly observed robust enrichments for specific sets of motifs in the promoters of all individual expression clusters. Three major classes emerge, within which motif enrichments are relatively homogeneous (*Figure 3B*).

Remarkably, these three classes define distinct temporal phases of embryonic development (*Figure 3A–B*). The Early expression class, enriched for DRE and Ohler-1/5/6/7 motifs, consists of

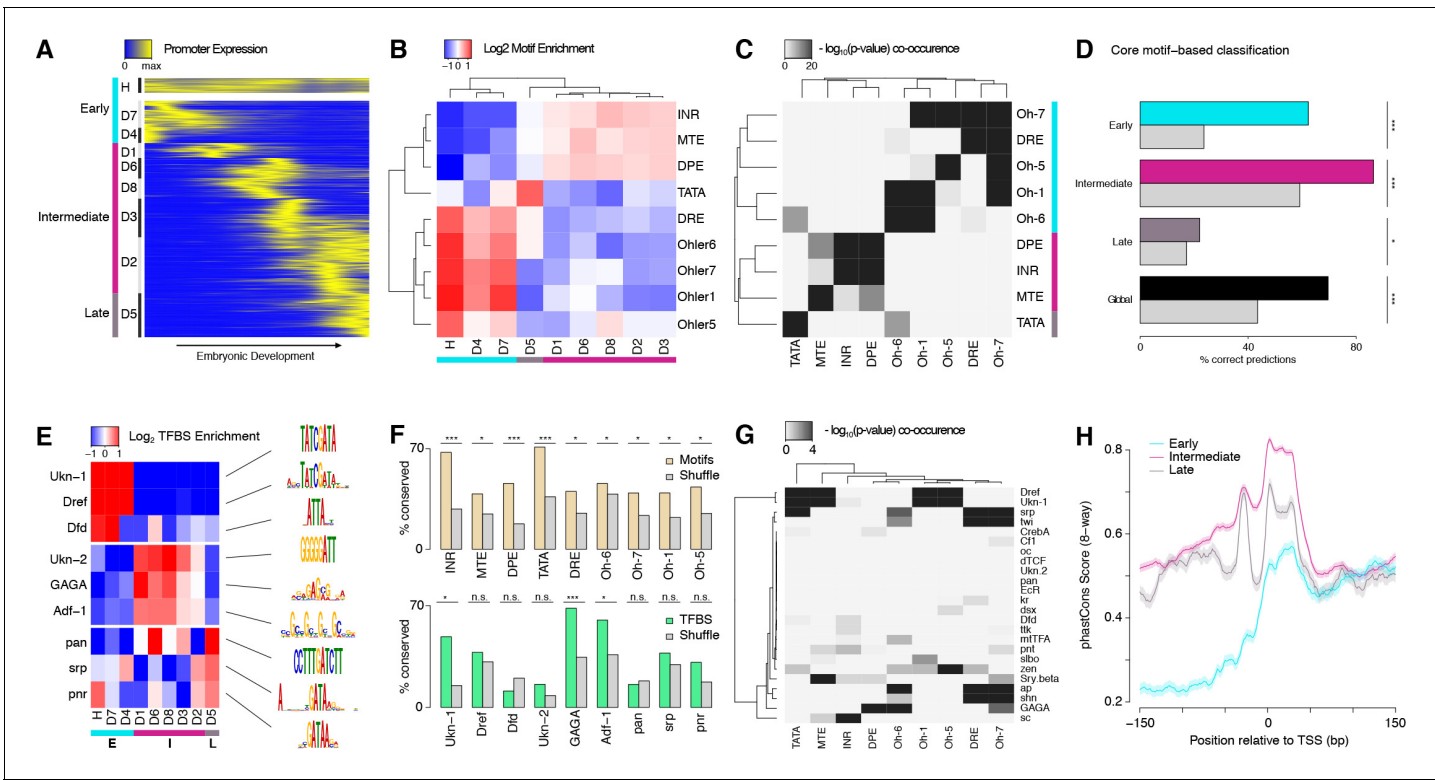

**Figure 3.** Core promoter structure defines three broad phases of embryonic development. (**A**) Clustering of expression profiles for 3462 promoters (maximum expression ≥10 RPM) defined as housekeeping (H, n = 240) or developmentally regulated (D, n = 3,222). Developmentally regulated promoters are divided into eight groups based on hierarchical clustering of expression profiles (see Materials and methods). Core promoter structure defines three broad developmental phases (color sidebar). (**B**) Relative enrichment of core promoter motifs in each expression cluster. Three major clusters can be defined (bottom color bar), which correspond to 3 phases of embryonic development (see (**A**); Early: 817 TSCs; Intermediate: 2,047; Late: 598). (**C**) Clustering of core promoter motifs based on their co-occurrence in the same promoters. This approach yields the same three major sets of motifs previously defined based on expression profiles (see (**B**)). (**D**) Predictions of expression timing from core promoter structure. Quadratic discriminant analysis on log-transformed motif scores was used to predict the developmental phase in which promoters are expressed. The performance of the classifier in leave-one-out cross-validation (color bars) is compared to random expectation (grey bars; FDR-corrected chi-square tests applied to individual contingency tables as appropriate). (**E**) Distinct sets of TFBSs are enriched near the core promoters active in the three developmental phases. The top three motifs for each core promoter class are shown. (**F**) The conservation of core promoter motif composition between *D. melanogaster* and *D. pseudoobscura* confirms the biological relevance of the motifs. (Grey bars: conservation of shuffled motifs; FDR-corrected chi-square tests). (**G**) Many TFBSs are strongly associated with specific sets of core promoter motifs. Results are shown for the three motifs most enriched near the promoters of each expression cluster. (**H**) The three major core promoter classes display distinct profiles of sequence conservation. Lines: median phastCons scores across promoters of the class. Envelope: standard error in the estimate of the median, computed by bootstrapping.

DOI: https://doi.org/10.7554/eLife.29005.010

The following figure supplement is available for figure 3:

**Figure supplement 1.** Core promoter types: Co-occurence of TFBSs and Sequence conservation profiles.
DOI: https://doi.org/10.7554/eLife.29005.011

the promoters for maternally deposited transcripts, including the housekeeping cluster. The Intermediate class, enriched for Initiator (INR), MTE and DPE motifs, mediates transcription throughout a broad phase of mid-embryogenesis, from the onset of zygotic expression to the end of organogenesis. The Late class, enriched for TATA boxes, drives transcription around the transition to the first larval stage. Notably, expression clusters with vastly different specificities (e.g., D1 and D2) share the same promoter structure trends.

Importantly, clustering core motifs by their tendency to co-occur within the same promoters, regardless of expression timing, recapitulates the same three main motif groups (*Figure 3C*). It does, however, also uncover a finer data structure, which suggests that there are a variety of promoter subtypes. In addition, we were able to train a classifier with substantial predictive power to distinguish the three promoter classes based solely on core motif scores (*Figure 3D*). Taken together, our observations show that specific sets of core promoter elements are preferentially associated with transcriptional activity during distinct phases of development. This suggests a possible mechanistic role for core promoter elements in defining windows of opportunity for promoter activity during distinct periods of development.

Sequences proximal to the three core promoter classes are also enriched for different sets of TFBS (*Figure 3E*). The Early class preferentially harbors binding sites for Dref and Dfd, while the Intermediate class favors Trl (GAGA motif) and Adf-1. The Late class is enriched in pan, srp and pnr sites. The presence of most core promoter motifs and TFBSs is conserved between species far beyond random expectations (*Figure 3F*), which validates the overall quality of our motif predictions. Interestingly, TFBSs are often specific for only a subset of expression clusters within a class (e.g., Dfd or GAGA), and tend to co-occur in stereotypical combinations, suggestive of widespread combinatorial encoding (*Figure 3—figure supplement 1*).

The analysis of favored pairings between individual TFBSs and core promoter motifs suggests a possible synergy in the specification of expression patterns. Indeed, some TFBSs appear to be strongly associated with specific sets of core promoter motifs (*Figure 3G*). Binding sites for the mesodermal factors srp and twi, for instance, which often tend to be found together (*Figure 3—figure supplement 1*), have a robust association with DRE and Ohler-7 core motifs. These strong affinities suggest that core motifs may tune the ability of a promoter to respond to specific sets of transcription factors. They may do so by recruiting different sets of general transcription factors (GTFs) that functionally interact with distinct groups of activators. Such a mechanism may channel various regulatory inputs to limited subsets of promoters and thus limit crosstalk between parallel pathways.

We found that the three promoter classes display markedly different profiles of sequence conservation (*Figure 3H* and *Figure 3—figure supplement 1*). Importantly, this analysis only includes those promoters for which we have detected transcriptional activity in all five species, and we can therefore categorically rule out differences in rates of promoter gain/loss as an explanation. These observations suggest that the three promoter classes indeed have intrinsic structural differences that make them subject to distinct regimes of natural selection and sequence evolution.

## Selection on expression specificity shapes promoter sequence evolution

To further explore the selective pressures acting on regulatory sequences, we investigated quantitative relationships between the evolution of promoter structure (primary sequence) and function (expression specificity). We report a subtle but highly significant correlation between the conservation of promoter sequence and that of expression profiles (*Figure 4A–B*). This shows that the effects of selection on expression specificity are reflected in the evolution of promoter sequences. The main areas of differential sequence conservation overlap regions preferentially occupied by precisely positioned core promoter elements (TATA, INR, DPE) and transcription factor binding sites (*Figure 4A*). Importantly, this correlation does not hold for sequences >50 bp downstream of the TSS, which are likely subject to additional selective pressures on 5′-UTR and protein-coding sequences. Promoters with highly divergent expression profiles are depleted of sites under purifying selection, and enriched for sites under positive selection (*Figure 4C–D*).

*Ab initio* motif discovery within the regions of differential sequence conservation returned the canonical consensus sequences for TATA, INR and DPE (*Figure 4E*). We found that promoters with highly conserved expression profiles tend to have core promoter elements whose sequence is closer to the motif consensus (*Figure 4F*), and those in turn tend to be more conserved at the sequence

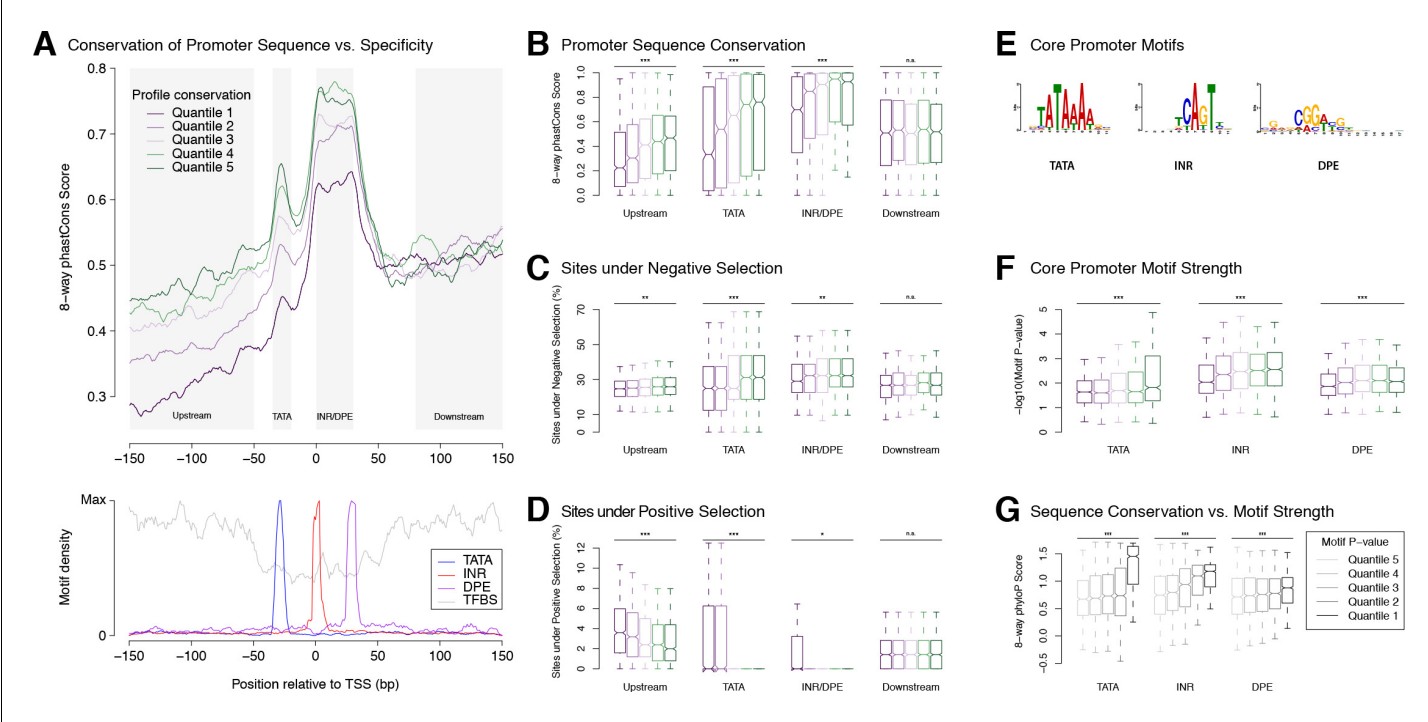

**Figure 4.** Selection on expression specificity shapes promoter sequence evolution. (**A**) Upper panel: average sequence conservation for *D. melanogaster* promoters, by expression profile conservation quantile. All *D. melanogaster* promoters with maximum expression ≥25 RPM and functionally conserved in all five species were included (n = 4,973). Lower panel: density of core promoter elements and TFBS over all promoters. TATA box, INR, DPE: respective consensus sequences STATAAA, TCAGTY or TCATTCG, KCGGTTSK or CGGACGT (*FitzGerald et al., 2006*); TFBS motifs from Jaspar. (**B**) Correlation between expression profile and promoter sequence conservation. For each of the subregions depicted as shaded areas in A (except Downstream), there is a significant correlation between expression profile conservation and phastCons score across all promoters analyzed (*** p-value<$10^{-8}$ for Pearson correlation test between profile conservation and sequence conservation; **p<$10^{-4}$; *p<$10^{-2}$; *n.s.* not significant. Upstream region runs from −300 to −50 bp). For graphical display, promoters were grouped into expression conservation quantiles, and the boxplots show the distribution of phastCons scores for each quantile. (**C**) Proportion of bases under purifying selection (phyloP score >0.1). (**D**) Proportion of bases under positive selection – that is, evolving faster than neutral sequences (phyloP score <−0.1) – at promoter subregions for each profile conservation quantile. (**E**) Core motif position weight matrices derived de novo from promoter sequences (see Materials and methods). (**F**) Core motif strength correlates with expression profile conservation. (**G**) Core motif sequence conservation correlates with proximity to motif consensus.

DOI: https://doi.org/10.7554/eLife.29005.012

The following figure supplement is available for figure 4:

**Figure supplement 1.** Conservation of transcription factor binding sites.
DOI: https://doi.org/10.7554/eLife.29005.013

level (*Figure 4G*). Upstream flanking sequences also tend to show higher conservation of individual TFBSs (*Figure 4—figure supplement 1*). Importantly, it is possible to detect such a correlation for the binding sites of some individual transcription factors (*Figure 4—figure supplement 1*). This is a rather striking fact, considering that promoter-proximal enhancers generally bind more than one factor, and that many promoters are additionally regulated by distal enhancers not taken into account here. Finally, the conservation of promoter TFBS composition, as expected, also correlates with interspecific divergence (*Figure 4—figure supplement 1*).

These observations establish a clear relationship between core promoter sequence and expression specificity. Selective constraints on developmental expression timing act particularly strongly on core promoter elements, most notably Initiator, DPE and TATA elements. This is consistent with the idea that some core promoter motifs could play a role as key determinants of developmental expression specificity. Together, our results lend support to the hypothesis that core motifs and general transcription factors play a crucial role in determining promoter expression specificity.

## Promoter birth and death are widespread and dynamic

In addition to changes in the specificity of individual promoters, gene expression programs evolve through turnover of regulatory elements (*He et al., 2011*; *Odom et al., 2007*; *Villar et al., 2015*). And indeed, we observed widespread birth and death of promoters throughout the clade: only 49% of *D. melanogaster* TSCs are functionally conserved in all five species (*Figure 5A*). To rule out genome assembly artifacts, we restricted our analysis to those with syntenic alignments to other genomes, and measured a functional conservation rate of 75% (*Figure 5A*). As some peaks lack alignments owing to genuine insertions or deletions, we expect the true conservation rate to be

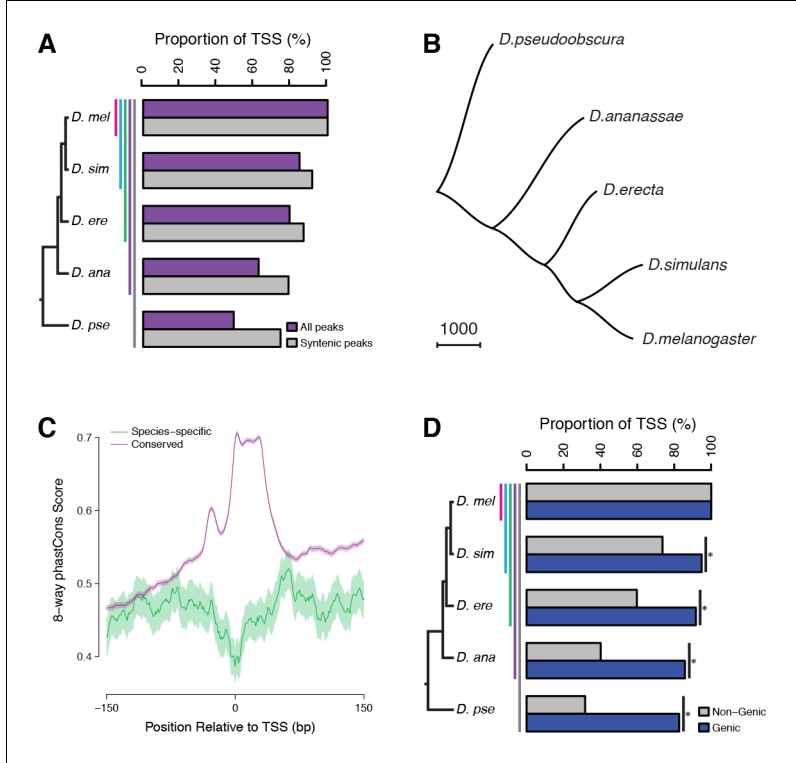

**Figure 5.** Widespread evolutionary gain and loss of promoters. (**A**) Proportion of *D. melanogaster* TSCs reproducibly detected in biological duplicates (first pair of bars) and functionally conserved in all species of subclades of increasing sizes. A TSC called in *D. melanogaster* Replicate 1 is deemed lost in another species (or replicate) if the orthologous region shows ≥100 fold reduced signal across the time course (see Materials and methods). Subclades include all descendants of a common ancestor, and are designated by the species that is most distantly related to *D. melanogaster*. (**B**) The species phylogeny can be accurately reconstructed from patterns of TSC gain and loss. The presence/absence of each TSC was treated as a discrete character and the unrooted tree reconstructed using the Phylip software package. (**C**) Average profiles of sequence conservation over the TSCs functionally conserved across all five species and those specific to *D. melanogaster*. The shaded areas represent the standard deviation, estimated from 1000 bootstraps. (**D**) TSCs driving the expression of annotated genes display a far higher degree of functional conservation than 'orphan' TSCs (p<0.01 for all pairwise comparisons; chi-square test with Bonferroni correction).

DOI: https://doi.org/10.7554/eLife.29005.014

The following figure supplements are available for figure 5:

**Figure supplement 1.** Alternative analyses of TSC conservation.
DOI: https://doi.org/10.7554/eLife.29005.015

**Figure supplement 2.** TSC conservation by expression quantiles.
DOI: https://doi.org/10.7554/eLife.29005.016

**Figure supplement 3.** Phylogeny of sequenced species.
DOI: https://doi.org/10.7554/eLife.29005.017

**Figure supplement 4.** Evolutionary rates of gain and loss for TSCs and Twist TFBSs.
DOI: https://doi.org/10.7554/eLife.29005.018

within the 49–75% range. Analyzing TSC conservation between species pairs, or from a *D. simulans*-centric perspective, yields similar conclusions (*Figure 5—figure supplement 1*).

The analysis of replicates for the *D. melanogaster* time series shows the false positive rate for gain/loss event detection to be under 0.1% (*Figure 5A*). Although TSCs with lower expression levels tend to be less conserved, general trends are shared between TSCs of all expression levels (*Figure 5—figure supplement 2*). Even though a gain/loss of expression during embryogenesis may reflect a shift in specificity rather than a complete gain/loss of function, the vast majority (91.4%) of *D. pseudoobscura* TSCs that were inferred to be lost in *D. melanogaster* based on embryo data are never expressed at any other stage of the life cycle (analysis of published data [*Batut et al., 2013*]). Therefore, we conclude that our strategy accurately and robustly detects promoter gain and loss events. Consistent with this, we can reconstruct the known species phylogeny by treating the presence or absence of individual TSCs as binary discrete characters in a standard parsimony framework (*Figure 5B*). Functional conservation is reflected in sequence conservation: promoters active in all species display far higher sequence conservation than species-specific ones, which appear no more constrained than surrounding regions (*Figure 5C* and *Figure 5—figure supplement 3*).

While purifying selection clearly plays a major role, a sizeable proportion of TSCs (25–50%) are not shared between all species, underscoring the inherently fluid nature of the regulatory landscape. Comparison to published data on the evolution of Twist binding sites (*He et al., 2011*) revealed that overall, promoters evolve no more slowly, and possibly even faster, than TFBSs do (*Figure 5—figure supplement 4*). Although a rigorous comparison between such disparate data types is delicate, this shows that promoters and TFBSs do not turn over on vastly different timescales.

Our data also reveals the existence of several thousand novel promoters that cannot be assigned to any annotated genes. Many of those may drive the expression of long noncoding transcripts, and it has long been a matter of debate to what extent this transcriptional activity plays meaningful biological roles. Hence we analyzed their rates of gain and loss to explore the selective pressures they may be subject to. We found a stark contrast in the degree of functional conservation of the two classes, with novel non-genic promoters evolving at a substantially higher rate than genic ones (*Figure 5D*). There is, however, a very substantial proportion of non-genic promoters that is deeply conserved (i.e., active in all five species analyzed), suggesting the possibility of widespread functionality. The discrepancy between the classes may be due to a larger proportion of noncoding transcripts being devoid of biological roles and evolving neutrally. Alternatively, it may instead reflect a more pronounced tendency for noncoding transcription to take on lineage-specific roles and thereby be a driver of adaptation, as has been suggested before. To gain a better understanding of these questions, we sought to investigate in more depth the evolution of noncoding transcription.

## Deep conservation of long noncoding RNA promoters

The prevalence and biological relevance of noncoding transcription have long been major areas of contention. Attempts at resolving these issues using genomic sequence conservation have been largely inconclusive, probably due to minimal selective constraints on the primary sequence of these transcripts (*Young et al., 2012*; *Derrien et al., 2012*; *Haerty and Ponting, 2013*). Studying promoter activity experimentally in a phylogenetic framework provides a unique opportunity to rigorously address the question of lncRNA conservation and functionality. Furthermore, our ability to pinpoint TSSs with single-nucleotide accuracy gives us unprecedented leverage to elicit elusive sequence conservation patterns. Furthermore, beyond sequence conservation, we are also in a position to assess selective constraints on the expression specificity of these promoters.

We found 3682 embryonic TSCs in *D. melanogaster* that could not be functionally linked to any annotated protein-coding or small RNA gene, and could therefore represent putative lncRNA promoters. We also identified TSCs for 291 annotated lncRNAs, bringing the total up to 3973. Their developmental expression kinetics appear to be diverse and exquisitely stage-specific (*Figure 6A*).

The analysis of published genome-wide DNaseI hypersensitivity data (*Thomas et al., 2011*) confirmed that these putative lncRNA TSCs are likely to correspond to genuine promoters (*Figure 6B* and *Figure 6—figure supplement 1*). In addition, to verify that the transcripts expressed from these TSCs are indeed independent and devoid of any significant protein-coding potential, we built transcript models from a recently published RNA-seq developmental time course (*Graveley et al., 2011*). We successfully generated transcript models for 16,105 TSCs, including 1475 lncRNA TSCs. Most of them appear to correspond to full-length transcripts, and the vast majority of putative

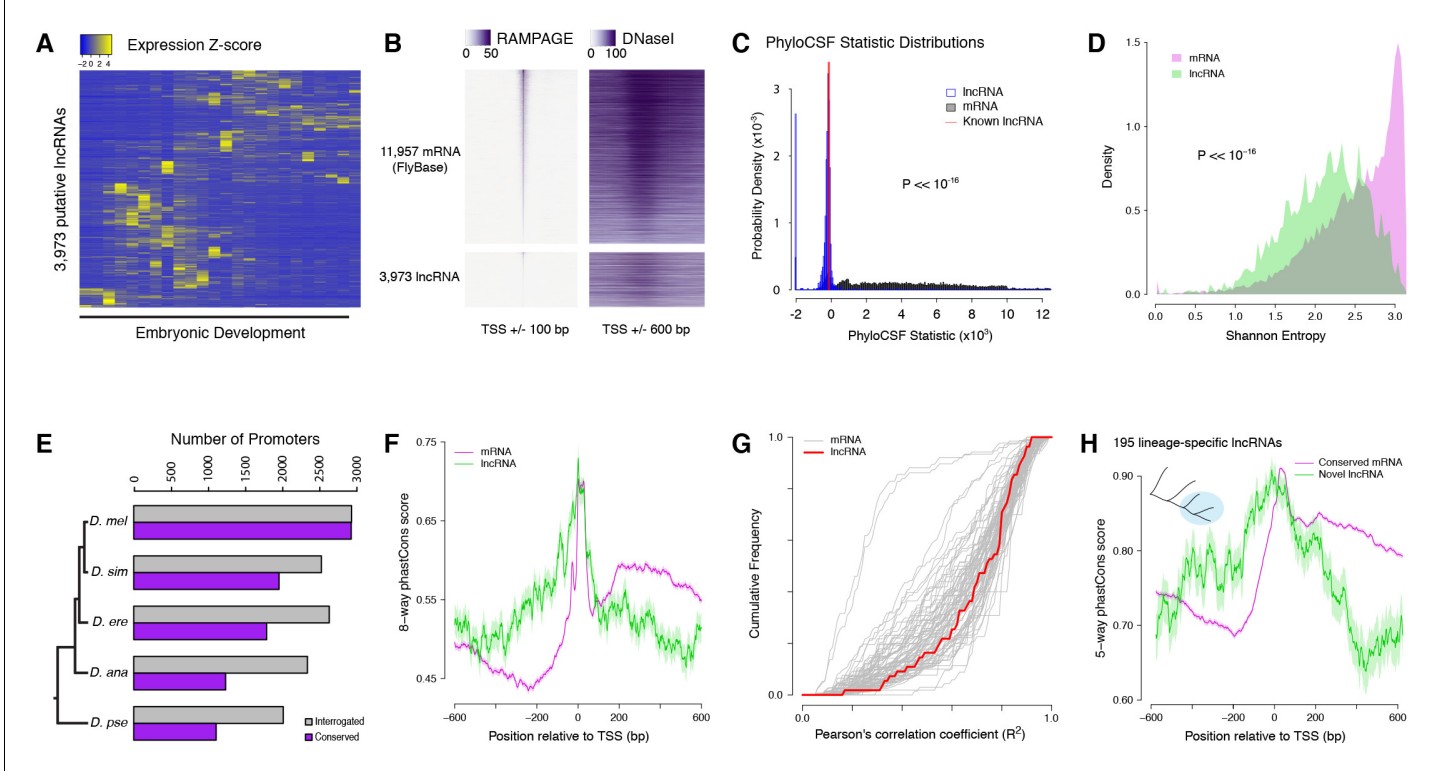

**Figure 6.** Strong purifying selection on long noncoding RNA promoters. (**A**) Developmental expression profiles of putative lncRNA promoters (n = 3,973). (**B**) Heatmaps of RAMPAGE signal (left; number of reads) and DNase-seq signal (right; arbitrary uits) over individual TSCs. We are comparing TSCs that overlap FlyBase-annotated mRNA transcription start sites (top), which we use as positive controls, to the TSCs of putative lncRNAs (bottom). In each group, TSCs are sorted by total RAMPAGE signal intensity. (**C**) Distribution of phyloCSF scores for transcript models corresponding to putative lncRNAs (n = 1,475) and mRNAs (n = 16,105). Transcript models were built from publicly available RNA-seq data using Cufflinks. The phyloCSF metric quantifies the protein-coding potential of transcripts, based on the presence and conservation of ORFs. (**D**) Shannon entropy of the temporal expression profiles for lncRNA (n = 2,397) and mRNA (n = 18,067) promoters with maximum expression ≥2 RPM. Overall, the profiles of lncRNA promoters have lower entropy, reflecting more acutely stage-specific expression. (**E**) Number of *D. melanogaster* lncRNA promoters functionally preserved in other species. The grey bars represent the number of promoters for which the multiple sequence alignments passed our filtering criteria, and therefore could be interrogated. (**F**) Promoter sequence conservation. Considering all promoters that are functionally preserved in all five species, the sequences of lncRNA promoters (n = 631) are under comparable selective pressure to those of protein-coding genes. (**G**) The developmental expression profiles of functionally conserved lncRNA promoters are far more constrained than those of many categories of protein-coding genes. We considered all promoters with maximum expression ≥25 RPM and expression changes ≥5 fold (n = 55 lncRNA promoters). (**H**) We identified 195 *D. melanogaster* lncRNA promoters that are functionally preserved within the *melanogaster* subgroup, but not in the two outgroup species, and are therefore likely to have been recently acquired specifically in this lineage (inset, top left). Lineage-specific lncRNA promoters display a level of sequence conservation within the subgroup similar to that of conserved protein-coding gene promoters.

DOI: https://doi.org/10.7554/eLife.29005.019

The following figure supplements are available for figure 6:

**Figure supplement 1.** DNase I hypersensitivity at RAMPAGE TSCs.
DOI: https://doi.org/10.7554/eLife.29005.020

**Figure supplement 2.** Independence and protein-coding potential of putative lncRNAs.
DOI: https://doi.org/10.7554/eLife.29005.021

**Figure supplement 3.** Clustering of *D. melanogaster* developmental expression profiles.
DOI: https://doi.org/10.7554/eLife.29005.022

**Figure supplement 4.** *Bithoraxoid* locus.
DOI: https://doi.org/10.7554/eLife.29005.023

**Figure supplement 5.** Conservation of *melanogaster* subgroup lncRNA TSCs.
DOI: https://doi.org/10.7554/eLife.29005.024

**Figure supplement 6.** Sequence conservation over *melanogaster* subgroup-specific lncRNA TSCs.
DOI: https://doi.org/10.7554/eLife.29005.025

lncRNAs do not overlap annotated protein-coding sequences (*Figure 6—figure supplement 2*). Analyses of protein-coding potential confirm that the overwhelming majority of transcripts are unlikely to encode proteins, or even peptides as short as 10 amino acids (*Figure 6C* and *Figure 6— figure supplement 2*). Transcripts from 18 loci are likely to encode short open reading frames (sORFs, <100 residues). We conclude that the vast majority of candidate transcripts are likely to be genuine lncRNAs.

The expression profiles of lncRNA promoters have unique properties that set them apart. As a class, they tend to be substantially more stage-specific than their protein-coding gene counterparts, as measured by the Shannon entropy of their activity profiles (*Figure 6D*). Comparing their developmental expression timing to that of protein-coding gene promoters suggests a broad diversity of potential developmental functions (*Figure 6—figure supplement 3* and *Supplementary file 1*).

The detection of lncRNA TSCs is highly reproducible across *D. melanogaster* biological replicates (*Figure 6E*). Of all *D. melanogaster* lncRNA TSCs, 2016 can be aligned to the *D. pseudoobscura* genome assembly and 1111 are functionally conserved (*Figure 6E*), suggesting that they have been maintained since the last common ancestor of these two species. In order to investigate whether lncRNA promoters are under purifying selection, we focused on an extremely stringently selected set of 631 TSCs that are active in all five species. This set includes well-known essential noncoding transcription units, such as *bithoraxoid* (*Lipshitz et al., 1987*; *Petruk et al., 2006*) (*Figure 6—figure supplement 4*) and *roX1* (*Franke and Baker, 1999*). Overall, the level of sequence conservation at these functionally preserved lncRNA promoters is similar to that observed at protein-coding gene promoters (*Figure 6F*). Their developmental expression specificity is also more constrained than that of many protein-coding gene promoters (*Figure 6G*). Both observations taken together argue strongly for sustained selective pressure on these 631 lncRNA promoters for 25–50 million years. Furthermore, many TSCs of interest were excluded from this analysis simply because of the poor quality of genome assemblies, and this is therefore an extremely conservative set. Therefore, to place a more reasonable lower bound on the true number of conserved lncRNA promoters, we focused on those shared between the 3 species of the *melanogaster* subgroup. These 1529 promoters similarly display a high degree of sequence conservation within the subgroup, and their expression specificity also appears constrained (*Figure 6—figure supplement 5*).

Still, it remains that lncRNA promoters as a class evolve much faster than those of protein-coding genes (*Figure 5D*), and it has been a matter of debate whether this reflects lineage-specific functions or merely neutral evolution. To address this question, we focused on 195 lncRNA TSCs that are specific to the *melanogaster* subgroup, despite the orthologous sequences being present in the genomes of the two outgroups (*Figure 6H*). Surprisingly, they display the same degree of sequence conservation as the protein-coding gene promoters that are shared throughout the subgroup (*Figure 6H*). The assessment of conservation at orthologous sequences in outgroup species confirms that the selective constraints are indeed lineage-specific (*Figure 6—figure supplement 6*). This argues that these evolutionarily recent lncRNAs have come under purifying selection after acquiring lineage-specific functions.

Taken together, our observations raise the possibility that a substantial proportion of lncRNAs may indeed be under purifying selection for biological functions relevant to embryonic development.

## *FBgn026449*: A deeply conserved, developmentally regulated lncRNA gene

To validate our findings in one specific case, we focused on the *FBgn0264479* locus (*Figure 7A*), which displays one the most tightly conserved expression patterns among all the lncRNA genes in our dataset. This is an intriguing embryonic transcript that, although it has been annotated based on expressed sequence tag (EST) data, has to our knowledge never been characterized. The 0.5 kb FBgn0264479 RNA is extremely unlikely to encode functional peptides, as assessed by phyloCSF analysis (score of −217.3) and manual curation (*Figure 7—figure supplement 1*). It is highly expressed in all five species surveyed, in a strikingly conserved temporal pattern restricted to a ~3 hr period encompassing the onset of gastrulation (*Figure 7B* and *Figure 7—figure supplement 2*). Northern-blot analysis confirmed the size and expression dynamics of the transcript (*Figure 7C* and *Figure 7—figure supplement 3*).

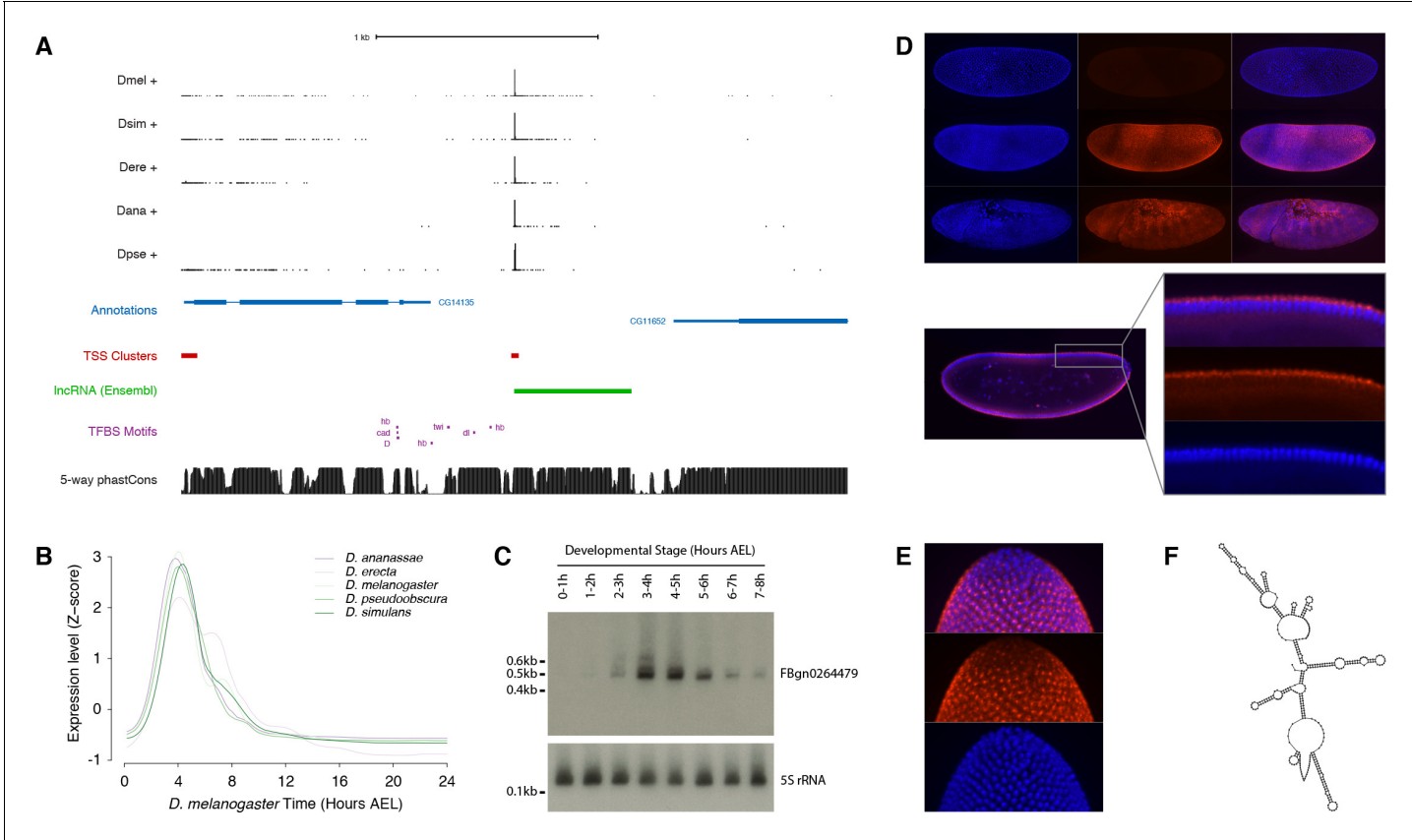

**Figure 7.** Functional characterization of the *FBgn0264479* locus. (**A**) Organization of the *FBgn0264479* locus (UCSC Genome Browser). Data tracks, from top to bottom: RAMPAGE signal in five species (five upper tracks), FlyBase gene annotations (blue), *D. melanogaster* TSCs (red), transcript annotation from Ensembl (green), TFBS motif predictions around the *FBgn0264479* promoter (purple), sequence conservation within the *melanogaster* subgroup (black). (**B**) FBgn0264479 transcript expression profiles in five species. (**C**) Northern-blot against the FBgn0264479 RNA. The 5S ribosomal RNA was used as a loading control (lower panel). (**D**) RNA-FISH for the FBgn0264479 transcript. Upper panel, top to bottom: maximum-intensity projections of confocal series for embryos at stages 4, 5 and 7 (Blue: DAPI, red: FISH). Bottom panel: Single confocal section from the embryo in the middle of the top panel. Controls with sense probes showed very little background. (**E**) FBgn0264479 RNA-FISH; lateral view of the posterior pole of a stage five embryo. (**F**) Mfold-predicted secondary structure of the RNA.

DOI: https://doi.org/10.7554/eLife.29005.026

The following figure supplements are available for figure 7:

**Figure supplement 1.** *FBgn0264479* protein-coding potential.
DOI: https://doi.org/10.7554/eLife.29005.027

**Figure supplement 2.** *FBgn0264479* locus and expression.
DOI: https://doi.org/10.7554/eLife.29005.028

**Figure supplement 3.** FBgn0264479 Northern-blot.
DOI: https://doi.org/10.7554/eLife.29005.029

**Figure supplement 4.** *FBgn0264479* transcriptional regulation.
DOI: https://doi.org/10.7554/eLife.29005.030

The body of the transcription unit displays hallmarks of robust purifying selection within the *melanogaster* subgroup (*Figure 7A*). In addition, publicly available chromatin immunoprecipitation data (*Li et al., 2008*; *MacArthur et al., 2009*) reveals the binding of several transcription factors to the promoter region (*Figure 7—figure supplement 2*), and their putative binding sites identified by sequence motif search also show evidence of purifying selection (*Figure 7—figure supplement 4*). The expression dynamics of the transcription factors are consistent with their regulating the *FBgn0264479* promoter (*Figure 7—figure supplement 4*).

Fluorescent in situ hybridization (FISH) in early embryos revealed expression along the ventral and dorsal midlines at the late blastoderm stage, with the exclusion of lateral regions, the primordial

germ cells and the prospective head (*Figure 7D*). This early expression domain subsequently evolves into a complex segmented pattern by the end of germband extension (*Figure 7D*). In the late blastoderm, the RNA is found almost exclusively in the cytoplasm at the apical pole of the cells (*Figure 7D*). It appears at that stage to be generally concentrated in a single major focus per cell (*Figure 7E*). Secondary structure predictions show that the FBgn0264479 transcript is likely to be highly structured (*Figure 7F*), suggesting a high potential for RNA-protein interactions.

Although further characterization will be required to decipher the precise biological role of this transcript, our observations definitively establish its developmental expression specificity, clearly point to a non-protein-coding function, and show that it has been under strong purifying selection over at least 25–50 million years.

## Discussion

This work provides, to our knowledge, the first genome-wide overview of promoter evolution in *Drosophila*, and we leveraged this comparative framework to study the sequence determinants of developmental expression specificity. Through nucleotide-resolution mapping of TSSs and quantitative measurements of expression kinetics, our study yields new insights into the transcriptional regulatory code. We find that distinct classes of core promoters drive transcription in three broad phases of embryonic development. Each class is defined by a characteristic set of core motifs, and is associated with regulation by specific groups of transcription factors. Of note, we successfully detected in vivo some functional associations between Dref and housekeeping promoters, and between Trl and some developmentally regulated promoters, that were recently described in vitro (*Zabidi et al., 2015*) (*Figure 3*). Our analysis generalizes the concept of specific interactions between core promoters and enhancer sequences, and demonstrates for the first time its global relevance in a developmental context.

We propose a hierarchical model for transcriptional regulation in which core promoter syntax defines broad temporal windows of opportunity for activation, and precise expression timing is subsequently refined by the binding of sequence-specific activators and repressors at enhancers. Core promoters may restrict regulatory inputs by recruiting different sets of general transcription factors (GTFs) that functionally interact with distinct groups of transcription factors (*Goodrich and Tjian, 2010*). Some GTFs have been shown to shape the expression specificity of individual promoters, and it is known that different activators and repressors have distinct requirements for cofactors and GTFs (*Zabidi et al., 2015*; *Goodrich and Tjian, 2010*). Such a mechanism may channel regulatory inputs to limited subsets of promoters, and thus limit crosstalk between promoters and enhancers across the genome. Notably, what applies to time may also apply to space, and it is possible that similar core promoter/enhancer interplay hierarchically specifies gene expression in broad developmental lineages and individual cell types.

Evolutionary analysis of developmental expression specificity further supports this model. First, the three major classes of core promoters defined here show drastically different patterns of sequence evolution, suggesting substantial differences in their underlying structure and functional interactions with the transcriptional machinery. At a finer scale, the conservation of expression specificity across species correlates with the degree of sequence conservation at canonical core promoter elements, such as TATA boxes and Initiator motifs. This is highly suggestive of an instructive role for these sequence elements, once thought generic, in defining developmental gene expression patterns.

Approximately 4000 promoters were found to drive the expression of lncRNAs during embryogenesis, a strikingly high number for this very brief developmental period. Our findings likely apply to other developmental stages as well, as we previously reported the existence of 7421 putative lncRNA promoters in an analysis of the whole *D. melanogaster* life cycle (*Batut et al., 2013*). In addition, we detected the expression of only 205 of 1119 recently identified lncRNAs (*Young et al., 2012*). This suggests that we are only beginning to scratch the surface of lncRNA biology in *Drosophila*. Importantly, we show here that vast numbers of these promoters are under strong selective pressure, at the levels of both promoter sequence and expression specificity. A *melanogaster* subgroup core set of at least 1529 is under substantial selective constraint, and most of those are therefore highly likely to have biologically relevant activities.

In agreement with previous reports (Derrien et al., 2012; Haerty and Ponting, 2013; Kutter et al., 2012), we find that lncRNA genes evolve faster than their protein-coding counterparts. It has been an unresolved debate so far whether this reflects neutral evolution or lineage-specific functions. Our observation that lineage-specific lncRNAs are also under substantial selective pressure reveals that noncoding transcription may be a major driver of phenotypic diversification and organismal adaptation. We recently showed that transposable elements play an important role in the evolutionary gain of promoters, and in particular of lncRNA promoters (Batut et al., 2013). We propose that transposon proliferation is a major mechanism favoring the neofunctionalization of intergenic regions as sources of biologically active noncoding transcripts.

To open a window into the biology of developmentally regulated lncRNAs, we focused on FBgn0264479, a deeply conserved yet never-before characterized gene. We experimentally validate the existence and expression kinetics of the FBgn0264479 transcript, and demonstrate that both its promoter sequence and its developmental expression specificity are deeply conserved across drosophilids. Interestingly, this lncRNA is expressed in a spatial profile highly similar to that of the schnurri gene (Arora et al., 1995), which is part of the Dpp (TGF-β/Smad) signaling pathway and plays an essential role in the establishment of dorsoventral embryo polarity (Arora et al., 1995; Dai et al., 2000). The punctate cytoplasmic localization pattern of the FBgn0264479 lncRNA is reminiscent of the targeting of multiple Dpp pathway components to endosomes in larval wing discs (Bökel et al., 2006; Coumailleau et al., 2009; González-Gaitán, 2003). Endosomes localize apically in the late embryonic blastoderm (Fabrowski et al., 2013), and mutants for Sara, the Smad endosome-targeting factor (Bökel et al., 2006; González-Gaitán, 2003), die early in embryogenesis (Bökel et al., 2006), suggesting that endosome-based signaling is also essential at that stage. This raises the intriguing possibility of a role for the FBgn0264479 lncRNA in TGF-β signaling, a crucial pathway in all animals that plays a major role in human disease, including cancer.

In recent years, it has become clear that noncoding transcription serves a myriad of molecular functions in Eukaryotes, and plays a part in virtually every known biological process (Augui et al., 2011; Ulitsky and Bartel, 2013; Guttman and Rinn, 2012; Ponting et al., 2009). LncRNAs have been shown to regulate transcription and chromatin structure, as well as mRNA stability and protein localization. Sometimes it is the transcription of the locus itself that plays a mechanistic role, rather than the resulting transcript – as in the case of the upstream bxd promoter (Petruk et al., 2006), which has one of the most highly conserved expression profiles that we have observed. Our work provides evolutionary evidence for a broad biological relevance of noncoding transcription to developmental processes, and establishes D. melanogaster as an excellent model for exploring the diverse functions of lncRNAs with targeted genetic and molecular studies. We expect that systematic efforts on a larger scale will illuminate the biology of a long-ignored class of genes that has proven its worth.

## Materials and methods

### Fly stocks and embryo collections

All Drosophila strains were obtained from the Drosophila Species Stock Center at UC San Diego, CA (https://stockcenter.ucsd.edu/info/welcome.php). For each species considered we worked with the reference genome strain. Stock numbers: D. melanogaster #14021–0231.36 (RRID:BDSC_2057), D. simulans #14021–0251.195 (RRID:FlyBase_FBst0201374), D. erecta #14021–0224.01 (RRID:FlyBase_FBst0200094), D. ananassae #14024–0371.13 (RRID:FlyBase_FBst0201380), D. pseudoobscura #14011–0121.94 (RRID:FlyBase_FBst0201265). Stocks were maintained on standard cornmeal medium. Embryo collections were performed in population cages (Flystuff, #59–116). 2- to 7-day-old flies were left to acclimatize to the cage for at least 48 hr and regularly fed with grape juice-agar plates (Flystuff, #47–102) generously loaded with yeast paste. After two 2 hr pre-lays, embryos were collected in 1 hr windows and aged appropriately (24 time points, 0–24 hr). Embryos were washed with deionized water, dechorionated for 90 s with 50% bleach, rinsed abundantly with water, and snap-frozen in liquid nitrogen.

## RNA extraction and RAMPAGE library preparation

Total RNA was extracted from embryos using a Beadbeater (Biospec, Cat. #607) with 1.0 mm zirconia beads (Biospec, #11079110zx) and the RNAdvance Tissue kit (Agencourt #A32649) according to the manufacturer's instructions, including DNaseI treatment. We systematically checked on a Bioanalyzer RNA Nano chip (Agilent) that the RNA was of very high quality. Libraries were prepared as described before (*Batut et al., 2013*; *Batut and Gingeras, 2013*). 5'-monophosphate transcripts were depleted by TEX digest (Epicentre #TER51020). For every time series, each sample was labeled with a different sequence barcode during reverse-transcription, and all samples for the series were then pooled and processed together as a single library. Quality control and library quantification were carried out on a Bioanalyzer DNA High Sensitivity chip. Each library was sequenced on one lane of an Illumina HiSeq 2000.

## Genome references and annotations

All reference sequences and annotations were obtained from Flybase (http://flybase.org). *D. melanogaster* release 5.49, *D. simulans* r1.4, *D. erecta* r1.3, *D. ananassae* r1.3, *D. pseudoobscura* r2.9.

## Primary data processing

Reads were mapped to the appropriate reference genomes using the STAR aligner (*Dobin et al., 2013*). Peaks were called on the pooled data from whole time series, using a custom peak-caller described previously (*Batut et al., 2013*; *Batut and Gingeras, 2013*). We used parameters optimized to yield good TSS specificity with respect to annotations and comparable numbers of peaks for all species. All peaks overlapping FlyBase-annotated rDNA repeats were filtered out.

## TSC conservation

Functional conservation was assessed for all peaks with $\geq$15 RAMPAGE tags that did not map to heterochromatic regions or chr4 in *D. melanogaster*, or orthologous regions in other species. We translated the genomic coordinates of each peak in each species to coordinates in the multiple sequence alignment of all genomes (15-way MultiZ alignment from UCSC, http://hgdownload.soe.ucsc.edu/goldenPath/dm3/multiz15way). To be considered for analysis, each peak was required to have a unique syntenic alignment in all other species considered, defined as follows: both ends of an 800 bp window centered on the middle of the peak had to map to the same strand of the same chromosome or scaffold, 50% of bases had to be aligned (i.e., not in assembly gaps), and 25% of bases had to align to orthologous bases (not alignment gaps). Raw 5' signal for each genome was also translated into multiple alignment coordinates. For each peak from each species, functional conservation was assessed by counting the number of RAMPAGE tags in each species. A peak was considered absent in a target species if it had at least a 100-fold lower signal than in the reference species. Peaks with <100 tags in the reference species were considered absent if they had no detectable signal in a target species.

## Phylogeny reconstruction

The peaks from all species were merged and collapsed in multiple alignment space to generate a non-redundant set of all peaks in the clade. The conservation of these peaks was assessed as described above. The phylogenetic tree was inferred by treating the presence/absence of each peak as a 2-state discrete character, sequentially using the MIX and PARS program of the PHYLIP suite according to the recommendations of the software documentation (http://evolution.genetics.washington.edu/phylip.html).

## Sequence conservation

Per-base conservation scores were computed by running the phastCons and phyloP programs of the PHAST suite v1.1 on the MultiZ alignment according to the recommendations of the software documentation (http://compgen.bscb.cornell.edu/phast). Depending on the subclade of interest, some species were excluded from the alignment for certain analyses. Pre-computed phastCons scores for the full 15-way alignment were downloaded from UCSC (http://hgdownload.soe.ucsc.edu/goldenPath/dm3/phastCons15way).

## Core promoter motifs

For analyses of motif composition, we only considered *D. melanogaster* TSCs that were functionally conserved across all five species. We used pairwise chained alignments downloaded from UCSC (http://hgdownload.soe.ucsc.edu/downloads.html#fruitfly) to align the most heavily used position of each TSC (i.e., the main TSS) to all other genomes. Peaks for which the maximum position could not be aligned to all genomes were excluded from the analysis. A custom script was used to search for matches to previously characterized core promoter motifs (*FitzGerald et al., 2006*) within a 301 bp window centered on the main TSS. Consensus sequences for sets of peaks with matches to individual motifs were computed using MEME v4.9.0 (http://meme.nbcr.net/meme).

## Time series alignment

Z-score transformed gene expression time series from all species were registered to one another using the GTEM suite (*Goltsev and Papatsenko, 2009*) according to the recommendations of the software documentation (http://flydev.berkeley.edu/cgi-bin/GTEM/index.html). One-to-one ortholy calls from Flybase (2012 release 2) were used to match gene expression profiles between species. We pre-processed pairs of datasets (*D. melanogaster* and another species) to compensate for differences in annotation quality and peak calling between species. We identified orthologs of TSCs that had detectable expression ($\geq$10 tags) but initially failed to be called in one species. In addition, when a functionally conserved TSC had been attributed to an annotated gene in one species but not the other, we corrected this discrepancy by attributing it to the gene in both species. For the *D. ananassae* dataset, the 8th time point failed to yield acceptable data, and was excluded from the analysis. All time series were upsampled 5-fold and smoothed with a 2 hr window size using RZ-Smooth v4.1. Optimal global alignment paths between *D. melanogaster* and the other datasets were computed with T-Warp v3.2 with Pearson distance matrices (3 hr window). M-Align v2.8 was used to align each series to the *D. melanogaster* reference and smooth the final aligned series (1 hr window). The expression profiles of individual TSCs were registered to one another with M-Align, using the optimal alignment path computed for gene expression profiles. Prior to alignment, we used the UCSC liftOver tool (http://hgdownload.cse.ucsc.edu/admin/exe/linux.x86_64/liftOver) to identify *D. melanogaster* TSCs that aligned well ($\geq$50% of bases aligned) to all other genomes. The temporal expression profiles of those orthologous genomic positions only were aligned.

## Expression profile conservation

We defined the conservation of individual expression profiles (TSCs or genes) across a clade as the average Pearson $R^2$ for all pairwise comparisons of species within the clade.

## Clustering of TSC expression profiles

We classified all *D. melanogaster* TSCs with maximum expression $\geq$10 RPM (n = 11,900) as either Housekeeping (<5 fold variation throughout the time series, n = 587) or Developmentally Regulated ($\geq$60% of total expression within an 8 hr window, n = 6,015). We further selected TSCs functionally conserved in all five species (n = 240 and n = 3,824, respectively). Developmentally regulated TSC profiles were hierarchically clustered (R *hclust*, distance metric *1 - cor(t(expr), method='pearson')*) and initially grouped into 12 clusters (R *cutree*, k = 12). After filtering out excessively small clusters (<200 TSCs, n = 4), further analysis was conducted on the remaining eight regulated clusters (n = 3,222 TSCs). See *Figure 3A* for clustering results.

## lncRNA transcript reconstruction and phyloCSF ORF analysis

We ran Cufflinks (v2.2.1) independently on each dataset of a published RNA-seq developmental series. Cuffmerge was used to generate a consensus annotation set. Transcript models were attributed to a RAMPAGE TSC if their 5' end lay within 150 of that TSC. Models without a matching TSC were excluded from further analyses. phyloCSF was run on these annotation sets and the 15-way multiZ whole-genome alignments.

## Analysis of sequence motifs

We used the MEME Suite v4.9.0 (primarily FIMO and MAST) to search promoter regions for a previously published compendium of motifs (*Stark et al., 2007*) corresponding to core promoter motifs

and TFBSs. MEME was used to generate consensus motifs from specific subregions of RAMPAGE-defined promoters (*Figure 4*).

## Other software

Custom analysis scripts were written in Python 2.7 (http://www.python.org). R was used for graphics generation (http://www.r-project.org).

### FBgn0264479 promoter sequence analysis

We identified potential regulators of the FBgn0264479 promoter based on the BDTNP transcription factor ChIP-chip data available from the UCSC genome browser website (*Li et al., 2008*; *MacArthur et al., 2009*). For 10 factors that bind the promoter in embryos (bcd, cad, D, dl, ftz, gt, h, hb, Kr, twi), we searched the 600 bp upstream of the main TSS for Jaspar TFBS motifs (http://jaspar.genereg.net), using FIMO. We identified seven motifs at a p-value cutoff of $2 \times 10^{-4}$.

## Northern-blot

We ran 8 μg of total RNA per sample on an 8% acrylamide 8M urea gel with a Invitrogen Novex minigel system. Transfer to a nylon membrane was carried out in 0.5X TBE in a Novex XCell II module, followed by UV-crosslinking (1,200J). For detection, we used a combination of 6 oligonucleotide probes targeting FBgn0264479 (5'-gaacatcgcttgcagtgcag, 5'-cgatggatgttgtcggtcgg, 5'-ctctcgttctttgattcttc, 5'-caggatgtgtggtgttccac, 5'-agattggatccttatggttg, 5'-atatgctgacactgcatggt). 30 pmol of oligo mix were radioactively labeled with γ (*Guttman and Rinn, 2012*)-ATP and PNK. Following phenol-chloroform extraction, the labeled probes were hybridized for 2 hr at 42°C in 40 mL of ULTRAHyb buffer (ThermoFischer). After serial washes with decreasing concentrations of SSC buffer (final stringency 0.5X), the membrane was exposed on Kodak BioMax autoradiography film. After stripping and control re-exposure, a similar protocol was used to detect the 5S rRNA on the same membrane, using a single probe (5'-caacacgcggtgttcccaagccg).

## Fluorescent in situ hybridization:

Templates for probe synthesis were generated by amplification of FBgn0264479 cDNAs with primers 5'-CGATGTTCTCCGACCGACAA and 5'-TGCACTACTTAGACTAAATTGGCT. In separate reactions, a T3 promoter sequence was added at either one end or the other, for the generation of sense and antisense probes. Amplicons were cloned into a TOPO-T/A vector (Life Technologies #K4575-01) and checked by Sanger sequencing. RNA-FISH was performed on 0–5 hr AEL *y; cn b sp* embryos as described before (*Legendre et al., 2013*), and imaged on a Perkin-Elmer UltraVIEW VoX confocal microscope. Biotin-conjugated mouse monoclonal anti-DIG (Jackson ImmunoResearch Laboratories Inc., Cat. No. 200-062-156) previously validated for this application (*Legendre et al., 2013*).

## Sample size estimates

In this work, relevant comparisons are between groups of promoters or groups of genes. All comparisons were designed to include all TSCs or genes of interest throughout the genome (e.g., all lncRNA TSCs *vs.* all genic TSCs), while applying expression level thresholds calibrated on the analysis of *D. melanogaster* replicates to ensure measurement reproducibility.

## Data availability

The primary data for this study is available through the GEO database, under accession numbers GSE36212 and GSE89335.

## Acknowledgements

The authors would like to thank Alexander Dobin, Felix Schlesinger, Chris Zaleski, Carrie Davis and all other members of the Gingeras group at CSHL for their assistance and advice, as well as Richard McCombie and the CSHL sequencing facility for their services. We also thank Alexander Gann, Gregory Hannon, Zachary Lippman, Joshua Dubnau, Adrian Krainer, Brenton Graveley, Jacques Batut and Mike Levine for helpful discussions and advice. We are grateful to Thomas Kaufman and the FlyBase

team for their permission to reproduce images. Work supported in part by the National Human Genome Research Institute, modENCODE Project, contract U01HG004271.

## Additional information

### Competing interests
Thomas R Gingeras: Reviewing editor, *eLife*. The other author declares that no competing interests exist.

### Funding

| Funder | Grant reference number | Author |
|---|---|---|
| Cold Spring Harbor Laboratory Watson School | Florence Gould Fellowship | Philippe J Batut |

The funders had no role in study design, data collection and interpretation, or the decision to submit the work for publication.

### Author contributions
Philippe J Batut, Conceptualization, Software, Formal analysis, Validation, Investigation, Visualization, Methodology, Writing—original draft; Thomas R Gingeras, Conceptualization, Resources, Supervision, Funding acquisition, Methodology, Project administration, Writing—review and editing

### Author ORCIDs
Philippe J Batut (iD) http://orcid.org/0000-0003-4250-0663
Thomas R Gingeras (iD) https://orcid.org/0000-0001-9106-3573

### Ethics
Animal experimentation: see attached upload.

### Decision letter and Author response
Decision letter https://doi.org/10.7554/eLife.29005.038
Author response https://doi.org/10.7554/eLife.29005.039

## Additional files

### Supplementary files
• Supplementary file 1. This table summarizes the most enriched Gene Ontology (GO) functional annotation categories for each gene expression cluster described in *Figure 6—figure supplement 3*.
DOI: https://doi.org/10.7554/eLife.29005.031
• Transparent reporting form
DOI: https://doi.org/10.7554/eLife.29005.032

### Major datasets
The following dataset was generated:

| Author(s) | Year | Dataset title | Dataset URL | Database, license, and accessibility information |
|---|---|---|---|---|
| Gingeras TR, Batut P | 2017 | Drosophila species embryonic development RAMPAGE time series | https://www.ncbi.nlm.nih.gov/geo/query/acc.cgi?acc=GSE89335 | Publicly available at the NCBI Gene Expression Omnibus (accession no. GSE89335) |

The following previously published datasets were used:

| Author(s) | Year | Dataset title | Dataset URL | Database, license, and accessibility information |
|---|---|---|---|---|
| Batut P, Dobin A, Plessy C, Carninci P, Gingeras TR | 2012 | Promoter activity profiling throughout the Drosophila life cycle reveals role of transposons in regulatory innovation | https://www.ncbi.nlm.nih.gov/geo/query/acc.cgi?acc=GSE36212 | Publicly available at the NCBI Gene Expression Omnibus (accession no. GSE36212) |

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
