## [Decision Letter]

Thank you for submitting your article "Conserved noncoding transcription and core promoter regulatory code in early *Drosophila* development" for consideration by *eLife*. Your article has been reviewed by three peer reviewers, one of whom is a member of our Board of Reviewing Editors, and the evaluation has been overseen by Didier Stainier as the Senior Editor. One of the three reviewers, James Bentley Brown, has agreed to reveal his identity.

The reviewers have discussed the reviews with one another and the Reviewing Editor has drafted this decision to help you prepare a revised submission.

Batut and Gingeras have collected and analyzed transcription start site data across the developmental time course for five *Drosophila* species. This large resource of transcription start site usage obtained at base resolution with a technology developed a few years ago by the authors (RAMPAGE). The results correspond to a temporal resolution of 1 hour of *Drosophila* development 5 species: *D. melanogaster, D. simulans, D. erecta, D. ananassae* and *D.*

*pseudoobscura* (120 samples). The analyses are based is a many as 2.7104 Transcription Start Clusters.

An analysis of the TSCs demonstrated that specific sets of core promoter sequence elements are associated with sets of TSCs that are activated at distinct times during development. This finding suggests that core promoter elements contribute to the temporal regulation of gene expression during differentiation.

The authors also reveal the presence of many TSCs that seem likely to be associated with lncRNAs and might be activated at a specific point in development.

Overall, the paper brings interesting resource to the transcription and *Drosophila* developmental communities that will contribute to the discussion of the role of core promoters in transcription regulation.

*Comments and required revision*

There are a number of aspects that require clarification. The most critical ones are points 1, 14, 17, 21, 24, 26, and 27. In general, the tone of some of the claims has to be moderated and adjusted to the actual observations.

1) The alignment of developmental profiles is crucial to the science that follows. Tried and true methods have been used to conduct the alignment, and it would also be good to generate a figure somehow visualizing the time-warping distance between the species across the developmental profile. I believe there was something similar to this between worm and fly in Gerstein et al. Nature. 2014 from modENCODE. You could also try improving on the modE graphic by making the connections between species' time-series heatmaps. You could arrange them in accordance with the phylogeny and then just show the pairwise mappings between neighboring species.

2) Figure 1, meant to be a graphical abstract, is confusing – it seems to relate a developmental profile directly to a specific genomic locus with arrows that point at the transcribed genome… very odd. I recommend refactoring to make it clear what was done. Or even just scrap it – the project description is very clearly written, not sure a graphical abstract is needed.

3) Figure 1 looks strange. Does it make sense to use the figure to show a single pick. In any case the figure requires a better introduction (Same for 1A)

4) Figure 1 is very beautiful, but may be over-smoothed – looks worse than the data actually is. It will be better re-doing the smoothing with a narrower bandwidth.

5) A better explanation of how clade and subclades are defined will help to understand the significance of Figure 2.

6) Expression profiles are overall tightly conserved across species (Figure 2 and Figure 2—figure supplement 4), but with substantial gene-to-gene variability:" It is not obvious from the figure.

7) The quantification of expression specificity in Figure 2, is peculiar. Will be helpful to show the error bars, number of cases involved and significance of the differences plotted in the figure.

8) The paragraph "The high similarity of biological replicates, the accuracy of inter-species alignments for well known developmental genes, and the biological features of evolutionary divergence patterns, together confirm our ability to accurately quantify promoter expression and its variation across species. Our observations also highlight the central importance of systems level selective constraints, such as those acting on gene function and developmental stages, in shaping the evolution of gene expression." Contains affirmation that I find difficult to follow in the data shown: accuracy of alignments for developmental genes? Importance of systems-level selective constraints? Evolution of gene expression?

9) Figure 3 The 3462 promoters where divided in housekeeping and developmentally regulated, but the number of developmentally regulated genes is again 3462 (817+2047+598), implying that the H genes are all early, but how many are they?

10) Are the same clusters shown in Figure 3? ("their tendency to co-occur 186 within the same promoters, regardless of expression timing, recapitulates the same three main motif groups (Figure 3).")

11) "Taken together, our observations suggest that core promoter elements play a significant role in restricting windows of opportunity for expression during distinct periods of development." Would not be more accurate to say that the analysis of the distribution of core promoters shows some differences in distribution than the general promoter expression? Or, otherwise, justify where the "restricting windows of opportunity for expression" has been deduced from?

12) Similar problems with the following affirmation "TFBSs are often specific for only a subset of expression clusters within a class (e.g., Dfd or GAGA). This suggests a model in which core promoter structure defines broad developmental periods of expression potential, and precise expression timing is then refined by sequence-specific transcription factors." This seems to be an anecdotal observation transformed in an important conclusion. Can the solid evidence supporting the very important claim on the general role of core promoters be pointed out clearly?

13) Figure 3., represents the conservation of core promoter motif between two species that substantiates a claim about conservation and validation of the motif prediction ("conserved between species far beyond random expectations (Figure 3), which validates the overall quality of our motif predictions") Why only two species, what happen with the clades and subclades? By the way, the font size is too small to be read in a print. Same for other figures too.

14) "Indeed, some TFBSs appear to be strongly associated with specific sets of core promoter motifs (Figure 3). Dref sites, for instance, are preferentially found along DRE and Ohler-1/6/7 core motifs." I cannot see this in the mentioned Figure 3. DRE is associated to oh1-oh5 mte and tata. I do not know if there is mistake in the text or my interpretation of the figure is wrong, but text and figure should be reconciled. In any case, the evidence presented in the figure seems to be very light to support the claim ("individual TFBSs and core promoter motifs suggests a possible mechanism to mediate this 2-step specification of expression patterns.")

15) In subsection “Global multispecies profiling of developmental promoter activity”: The GO term analysis seems to suggest that differential rates of cell proliferation across the 5 species may drive the signal, which would be consistent with previous modENCODE results. Any chance that you already have data from which you could quantify rates of cell duplication, or even just look at approximate cell counts and then do a simple regression to confirm? No pressure, but would put this one to rest once and for all.

16) The clustering analysis is somewhat reminiscent of promoter structure analysis in zebrafish. Might be good to cite: https://www.ncbi.nlm.nih.gov/pmc/articles/PMC4820030/

17) Figure 4 is key. The analysis by profile conservation quantile seems a little be of a strange choice, where the rest of the paper is based on continuous measurements. It will be good to read the justification of the authors on this point.

18) Figure 4. The positive selection panel (4D) is really not very demonstrative, is it? In any case, it needs a better explanation.

19) The results are show tendencies for three promoter elements: Initiator, DPE and TATA elements. Is justify to generalize to others ("This is consistent with the idea that core promoter motifs are indeed key determinants of this specificity")?

20) In subsection “Promoter birth and death are widespread and dynamic”; the statement that Figure 5 shows that the FDR on promoter gain/loss events is less than 0.1% is not clear, from which data does this follow? Please clarify.

21) The second part of the papers deals with the analysis of non-genic promoters. "We found a stark contrast in the degree of functional conservation of the two classes, with novel non genic promoters evolving at a substantially higher rate than genic ones (Figure 5). There is, however, a very substantial proportion that is deeply conserved, suggesting the possibility of widespread functionality" Given the importance of this observation some quantification will be interesting. Is the 20% of non-genic promoters conserved between D. pse and D. mel, what the authors understand by "deeply conserved". How many of the 3682 non-genic promoters are "deeply" conserved? Is there an analysis of those promoters in comparison with the genic ones? How similar or different are they from the additional 291 promoters of lncRNAs added to the analysis?

22) It follows with the key question of this second part of the paper, namely: "The discrepancy between the classes may be due to a larger proportion of noncoding transcripts being devoid of biological roles and evolving neutrally. Alternatively, it may instead reflect a more pronounced tendency for noncoding transcription to take on lineage-specific roles and thereby be a driver of adaptation, as has been suggested before."

23) The analysis of the expression reveals complex heterogeneous patters that are positively interpreted as representing possible roles of the lncRNAs, obviously following the paragraph above the opposite interpretation is equally plausible. The analysis of conservation across species has to be the central argument in this paper and this is what is proposed: "Of all *D. melanogaster* lncRNA TSCs, 2,016 can be aligned to the D. pseudoobscura genome assembly and 1,111 are functionally conserved" (what is functionally conserved in this context?). Does this information solve the question? Is 1111 out of 2016 significant? Is the analysis of the 631 or 1529 promoters conserved across species the best solution? Would not be better a pair-wise comparison between the five species? (All what does not detract from the interesting findings regarding the conservation of the properties of the set of 631 conserved promoters of lncRNAs).

24) What is less convincing is the final argument based on the analysis of the lncRNA TSCs that are specific to the *melanogaster* subgroup based the results shown in Figure 6, that shows some similarity between with the conservation profiles of protein coding gene promoters. It seems to be a little be exaggerated, based on the data presented, to conclude "Taken together, our observations show that a vast proportion of lncRNAs are indeed under purifying selection for biological functions relevant to embryonic development."

25) The arguments on the role of the lncRNA TSCs, based on conservation of a subset of them, are not sufficiently strong to close the debate on the actual role of the lncRNA, even if they may serve to point to the function of some specific cases in development control. Along this line, the argument in the final paragraph of the Discussion section, that the work unambiguously demonstrates the relevance of lncRNAs to development is also a bit strong. The work is highly suggestive, but careful knockouts or other perturbation experiments would be needed to "unambiguously demonstrate" – and even then proving that it isn't actually some short ORF would be tough. I recommend softening this language a bit to avoid raising the hackles of the lncRNA and/or the smORF communities.

26) In the third paragraph of subsection “schnurri-like RNA: A deeply conserved, developmentally regulated lncRNA gene”; the claim that the expression pattern is similar to shn seems odd. The gene shn is expressed throughout much of development (certainly through stage 16), it is not restricted to a 3 hour period as with this lncRNA. Here are the in situs of shn: http://insitu.fruitfly.org/cgi-bin/ex/report.pl?ftype=1&ftext=FBgn0003396. Do the authors mean just at this developmental period? If so, a comparison to many other expression profiles, e.g. from the BDGP, would be needed to support the association between this lncRNA and shn – is shn really the "most" similar at this period?

27) Further, the statements in the final paragraph of subsection “schnurri-like RNA: A deeply conserved, developmentally regulated lncRNA gene” that the authors have "confirmed it's noncoding nature" is too bold. The data is consistent with a non-coding transcript, but it is far from "confirmed". Absence of evidence is not necessarily evidence of absence. And on the contrary, doesn't it look quite cytoplasmic in the FISH? It encodes several ORFs > 20aa – I recommend softening this statement a bit. It looks non-coding, it's probably non-coding, but it could also encode a short ORF – to go the extra mile on this would require targeted quantitative proteomics. Not necessary for publication of course, but without it, the statement should be softened.

28) The TGF-β link is also quite a leap – it would be good to strengthen this portion of the paper with a bioinformatics analysis of the in situ imaging data against the BDGP database.

29) The observation that specific core promoter elements are associated with TSCs that are activated at specific points during development was made using the subset of TSCs that are functionally active in all species. If the analysis is performed on all TSCs observed in *Drosophila melanogaster*, are the same trends observed? Or is this observation specific to conserved TSCs?

30) Are the same core promoter element/TFBS trends seen for the non-coding TSCs? In other words, do you see a relationship between the promoter sequence elements for non-coding TSCs and their timing of activation?

---

## [Author Response]

1) The alignment of developmental profiles is crucial to the science that follows. Tried and true methods have been used to conduct the alignment, and it would also be good to generate a figure somehow visualizing the time-warping distance between the species across the developmental profile. I believe there was something similar to this between worm and fly in Gerstein et al. Nature. 2014 from modENCODE. You could also try improving on the modE graphic by making the connections between species' time-series heatmaps. You could arrange them in accordance with the phylogeny and then just show the pairwise mappings between neighboring species.

The alignment of the time series is indeed central to much of the analysis that follows. We agree that readers would benefit from more information on this process. We have incorporated into the supplementary material a new figure that shows the alignment paths for all pairwise global alignments to *D. melanogaster* (Figure 2—figure supplement 2).

2) Figure 1, meant to be a graphical abstract, is confusing – it seems to relate a developmental profile directly to a specific genomic locus with arrows that point at the transcribed genome… very odd. I recommend refactoring to make it clear what was done. Or even just scrap it – the project description is very clearly written, not sure a graphical abstract is needed.

This figure has been modified in the revised manuscript. The legend has been edited as follows: “(a) Genome-wide TSS usage maps were generated by RAMPAGE assays for developmental series in 5 species. These datasets constitute single-nucleotide measurements of transcriptional activity at 1-hour resolution throughout embryonic development, and a global survey of its evolution over 2550MY.

3) Figure 1 looks strange. Does it make sense to use the figure to show a single pick. In any case the figure requires a better introduction (Same for 1A)

Since there is a chance that this type of transcriptome data is unfamiliar to many readers, we feel that a brief graphical description of its nature, along with some salient features of the analysis, might help bring some clarity. The figure panel was reduced in size so as to appear less prominent, and the legend was rewritten as follows:

“(b) This panel illustrates the 5-species RAMPAGE data at a conserved locus (*NLaz*). The top 5 tracks show the density of RAMPAGE read 5’ ends over the locus. The non-*melanogaster* data was aligned to the appropriate genome references and projected for visualization onto the orthologous positions in *D. melanogaster*. The next tracks show the Transcription Start Site Cluster (TSC) called from this 5’ end data in *D. melanogaster*, and the partial transcript model reconstructed for this TSC from the full sequencing reads. Note the agreement with the transcript annotated in FlyBase, in blue, for the portion covered by the data. The last track shows phastCons sequence conservation scores.”

4) Figure 1 is very beautiful, but may be over-smoothed – looks worse than the data actually is. It will be better re-doing the smoothing with a narrower bandwidth.

This figure depicts the post-alignment cross-correlation between the two *D. melanogaster* time series, and reflects the temporal smoothing of gene expression profiles that is part of the preprocessing of the data. This used the same up sampling and Gaussian smoothing parameters (as recommended by the developers of the algorithm, see Materials and methods) as for the profiles depicted in Figure 2. However, we did use a color scale for the heatmap that indeed makes it appear unnecessarily blurry. We have rectified the color scale in the new version of Figure 1.

5) A better explanation of how clade and subclades are defined will help to understand the significance of Figure 2.

For more clarity, the legend for Figure 2 was rewritten as follows:

“(c) Conservation of the temporal expression profiles of individual promoters for all subclades of the phylogeny (subclades include all species descended from a common ancestor, and are coded by color bars next to the tree). For each subclade, we computed the average correlation coefficient between all pairs of species for each individual gene. The graph shows the median and first quartile over all genes with orthologs throughout the subclade.”

6) Expression profiles are overall tightly conserved across species (Figure 2 and Figure 2—figure supplement 4), but with substantial gene-to-gene variability:" It is not obvious from the figure.

Figure 2 shows what we consider to be an overall strong conservation of gene expression profiles: median R^2^~0.8 genome-wide for all 5 species, over 25-50MY of divergence. Figure 2—figure supplement 4 shows, however, that not all genes are equal with regards to the conservation of their expression profiles: the full distributions show long tails reaching down to very low correlation coefficients. Figure 2—figure supplement 4 and 4C show examples of genes with highly and poorly conserved profiles, respectively. We have not been able to find a compelling way to do this, but would be happy to incorporate any suggestions so as to make this more apparent from the main figure alone. The text has also been modified as follows:

“For all genes with one-to-one orthologs, the expression profiles are overall tightly conserved across species (Figure 2), but with substantial gene-to-gene variability (Figure 2—figure supplement 6A)”

7) The quantification of expression specificity in Figure 2, is peculiar. Will be helpful to show the error bars, number of cases involved and significance of the differences plotted in the figure.

Figure 2 has been modified to incorporate error bars. The legend has been edited as follows: “The evolutionary divergence of expression specificity varies widely between Gene Ontology (GO) categories (p<10^-16^, one-way ANOVA). For each gene with a GO annotation, orthologs in all 5 species, maximum expression ≥25 RPM and expression changes ≥5-fold (n=2,690), we computed a measure of overall divergence across the clade (see Materials and methods). The bar plot shows the average divergence by GO category, for the 20 categories with the lowest (top) and greatest (bottom) divergence. Error bars show +/-1 standard deviation. “

8) The paragraph "The high similarity of biological replicates, the accuracy of inter-species alignments for well known developmental genes, and the biological features of evolutionary divergence patterns, together confirm our ability to accurately quantify promoter expression and its variation across species. Our observations also highlight the central importance of systems level selective constraints, such as those acting on gene function and developmental stages, in shaping the evolution of gene expression." Contains affirmation that I find difficult to follow in the data shown: accuracy of alignments for developmental genes? Importance of systems-level selective constraints? Evolution of gene expression?

The first sentence simply recapitulates the observations made in the previous paragraphs: “high similarity of biological replicates”: Figure 1 “accuracy of inter-species alignments for well-known developmental genes”: Figure 2 “biological features of evolutionary divergence patterns”: Figure 2

The second sentence is indeed somewhat confusing. This paragraph has been revised as follows: “The high similarity of biological replicates, the accuracy of inter-species alignments for well-known developmental genes, and the biological features of evolutionary divergence patterns, together confirm our ability to accurately quantify promoter expression and its variation across species. Our observations also highlight the role of systems-level selective constraints, such as those acting on particular functional gene classes (Figure 2), in shaping the evolution of gene expression.”

9) Figure 3 The 3462 promoters where divided in housekeeping and developmentally regulated, but the number of developmentally regulated genes is again 3462 (817+2047+598), implying that the H genes are all early, but how many are they?

As defined by our filtering criteria, we are considering n=3,222 developmentally regulated promoters, and n=240 defined as housekeeping (further details in Materials and methods section). For subsequent analysis, the housekeeping promoters were indeed grouped together with the early developmental genes based on their sequence motif composition (Figure 3). The figure legend was modified to include this information.

10) Are the same clusters shown in Figure 3? ("their tendency to co-occur 186 within the same promoters, regardless of expression timing, recapitulates the same three main motif groups (Figure 3).")

Figure 3 do end up showing the same motif clusters, but not by design – this is actually a somewhat interesting result. Analyzing sequence motif enrichments in the promoters previously clustered by expression profile (Figure 3) reveals particular motifs associated with the 3 expression classes (Figure 3). An independent analysis of motif co-occurrence within individual promoters, uninformed by developmental expression data, yields the same groups of motifs (INR/MTE/DPE, DRE/Ohler-1/5/6/7, and TATA). This confirms the conclusion drawn from Figure 3 that there are indeed 3 major classes of promoters that are distinguished by their motif composition. This is further corroborated by the predictive power of motif composition in classifying promoters into the 3 main expression classes (Figure 3).

11) "Taken together, our observations suggest that core promoter elements play a significant role in restricting windows of opportunity for expression during distinct periods of development." Would not be more accurate to say that the analysis of the distribution of core promoters shows some differences in distribution than the general promoter expression? Or, otherwise, justify where the "restricting windows of opportunity for expression" has been deduced from?

This passage has been modified as follows:

“Taken together, our observations show that specific sets of core promoter elements are preferentially associated with transcriptional activity during distinct phases of development. This suggests a possible mechanistic role for core promoter elements in defining windows of opportunity for promoter activity during distinct periods of development.”

12) Similar problems with the following affirmation "TFBSs are often specific for only a subset of expression clusters within a class (e.g., Dfd or GAGA).This suggests a model in which core promoter structure defines broad developmental periods of expression potential, and precise expression timing is then refined by sequence-specific transcription factors." This seems to be an anecdotal observation transformed in an important conclusion. Can the solid evidence supporting the very important claim on the general role of core promoters be pointed out clearly?

We agree and thus for clarity and balance, speculation on mechanistic interpretations of our findings has been moved to the Discussion section.

13) Figure 3., represents the conservation of core promoter motif between two species that substantiates a claim about conservation and validation of the motif prediction ("conserved between species far beyond random expectations (Figure 3), which validates the overall quality of our motif predictions") Why only two species, what happen with the clades and subclades? By the way, the font size is too small to be read in a print. Same for other figures too.

This analysis is intended merely as a technical validation of our motif predictions, rather than as a basis for any interesting statement about patterns of promoter evolution. Figure 3 shows that hits for the motifs of interest are more conserved that hits for shuffled versions of themselves, which suggests that our predictions are enriched for biologically relevant sequences. The use of the 2 most distant species was just a rather obvious choice for this analysis, because it maximizes the sequence divergence for the regions analyzed.

As suggested, we have also modified the font size in figures.

14) "Indeed, some TFBSs appear to be strongly associated with specific sets of core promoter motifs (Figure 3). Dref sites, for instance, are preferentially found along DRE and Ohler-1/6/7 core motifs." I cannot see this in the mentioned Figure 3. DRE is associated to oh1-oh5 mte and tata. I do not know if there is mistake in the text or my interpretation of the figure is wrong, but text and figure should be reconciled. In any case, the evidence presented in the figure seems to be very light to support the claim ("individual TFBSs and core promoter motifs suggests a possible mechanism to mediate this 2-step specification of expression patterns.").

We thank the reviewer for catching this. This was indeed a mistake in the text. This passage has been edited and condensed as follows:

“Indeed, some TFBSs appear to be strongly associated with specific sets of core promoter motifs (Figure 3). Binding sites for the mesodermal factors srp and twi, for instance, which often tend to be found together (Figure 2—figure supplement 4), have a robust association with DRE and Ohler-7 core motifs.”

Here as well, we have moved this conjecture to the Discussion section.

15) In subsection “Global multispecies profiling of developmental promoter activity”: The GO term analysis seems to suggest that differential rates of cell proliferation across the 5 species may drive the signal, which would be consistent with previous modENCODE results. Any chance that you already have data from which you could quantify rates of cell duplication, or even just look at approximate cell counts and then do a simple regression to confirm? No pressure, but would put this one to rest once and for all.

Unfortunately, we are not aware of any existing datasets that would allow us to conduct this analysis.

16) The clustering analysis is somewhat reminiscent of promoter structure analysis in zebrafish. Might be good to cite: https://www.ncbi.nlm.nih.gov/pmc/articles/PMC4820030/

We have included this reference as suggested.

17) Figure 4 is key. The analysis by profile conservation quantile seems a little be of a strange choice, where the rest of the paper is based on continuous measurements. It will be good to read the justification of the authors on this point.

This was indeed not clear from the text, but the correlations between sequence and expression profile conservation were actually computed in a promoter-wise fashion, using continuous measurements. The grouping by quantiles was used only for display purposes, as it does a better job than scatterplots at showing these subtle correlations. All correlation tests were conducted on the full population of promoters, regardless of quantile groupings, and the p-values in this figure reflect these analyses.

For clarity, the figure legend has been edited as follows:

“(b) Correlation between expression profile and promoter sequence conservation. For each of the subregions depicted as shaded areas in A (except Downstream), there is a significant correlation between expression profile conservation and phastCons score across all promoters analyzed (*** pvalue < 10^-8^ for Pearson correlation testbetween profile conservation and sequence conservation; ** p < 10^-4^; * p < 10^-2^; *n.s.* not significant; Upstream region runs from -300 to -50bp). For graphical display, promoters were grouped into expression conservation quantiles, and the boxplots show the distribution of phastCons scores for each quantile.”

18) Figure 4. The positive selection panel (4d) is really not very demonstrative, is it? In any case, it needs a better explanation.

Figure 4 offer complementary analyses of the distinct roles of negative (purifying) and positive (diversifying) selection in shaping promoter evolution.

Figure 4 shows a positive correlation between the conservation of expression profiles and the number of sites under purifying selection. This suggests that purifying selection plays a significant role in constraining promoter expression divergence.

As might have been expected, Figure 4 shows a much more modest, yet significant, negative association with positive selection as well. This is particularly clear for the Upstream subregion, which is the largest one and thereby offers the most statistical power for this analysis. This observation at least raises the possibility that there may be a subpopulation of fast-evolving promoters whose expression divergence is driven by diversifying selection, which seems worth noting.

The figure legend has been edited as follows:

“(d) Proportion of bases under positive selection – *i.e.*, evolving faster than neutral sequences (phyloP score <-0.1) – at promoter subregions for each profile conservation quantile.”

19) The results are show tendencies for three promoter elements: Initiator, DPE and TATA elements. Is justify to generalize to others ("This is consistent with the idea that core promoter motifs are indeed key determinants of this specificity")?

Here we are not attempting to formally generalize this conclusion to all core promoter motifs: we’re merely raising the possibility that core promoter motifs – at least some of them – may shape promoter expression specificity.

This sentence has been edited as follows:

“This is consistent with the idea that some core promoter motifs could play a role as key determinants of developmental expression specificity.”

20) In subsection “Promoter birth and death are widespread and dynamic”; the statement that Figure 5 shows that the FDR on promoter gain/loss events is less than 0.1% is not clear, from which data does this follow? Please clarify.

The procedure to assess TSC functional conservation is described in detail in the Materials and methods section, in the “TSC conservation” subsection.

However, for clarity, the figure legend has been edited as follows:

“(a) Proportion of *D. melanogaster* TSCs reproducibly detected in biological duplicates (first pair of bars) and functionally conserved in all species of subclades of increasing sizes. A TSC called in *D. melanogaster* Replicate 1 is deemed lost in another species (or replicate) if the orthologous region shows ≥100-fold reduced signal across the time course (see Materials and methods). Subclades include all descendants of a common ancestor, and are designated by the species that is most distantly related to *D. melanogaster*.”

21) The second part of the papers deals with the analysis of non-genic promoters. "We found a stark contrast in the degree of functional conservation of the two classes, with novel non genic promoters evolving at a substantially higher rate than genic ones (Figure 5). There is, however, a very substantial proportion that is deeply conserved, suggesting the possibility of widespread functionality" Given the importance of this observation some quantification will be interesting. Is the 20% of non-genic promoters conserved between D. pse and D. mel, what the authors understand by "deeply conserved". How many of the 3682 non-genic promoters are "deeply" conserved? Is there an analysis of those promoters in comparison with the genic ones? How similar or different are they from the additional 291 promoters of lncRNAs added to the analysis?

By “deeply conserved”, we mean promoters that are functionally conserved (see Materials and methods, “TSC conservation”) in all 5 species analyzed. The sentence above has been edited to clarify this: “There is, however, a very substantial proportion of non-genic promoters that is deeply conserved

(*i.e*., active in all 5 species analyzed), suggesting the possibility of widespread functionality. “Roughly a third of all *D. melanogaster* non-genic TSCs are “deeply conserved” by this definition (Figure 5). Figure 6 presents multiple comparative analyses of genic *vs*. non-genic promoters and the transcripts that are produced from them.

22) It follows with the key question of this second part of the paper, namely: "The discrepancy between the classes may be due to a larger proportion of noncoding transcripts being devoid of biological roles and evolving neutrally. Alternatively, it may instead reflect a more pronounced tendency for noncoding transcription to take on lineage-specific roles and thereby be a driver of adaptation, as has been suggested before."

This passage simply puts forward the 2 alternative interpretations for this observation, which are further investigated in what follows in the next section.

23) The analysis of the expression reveals complex heterogeneous patters that are positively interpreted as representing possible roles of the lncRNAs, obviously following the paragraph above the opposite interpretation is equally plausible.The analysis of conservation across species has to be the central argument in this paper and this is what is proposed: "Of all D. melanogaster lncRNA TSCs, 2,016 can be aligned to the D. pseudoobscura genome assembly and 1,111 are functionally conserved" (what is functionally conserved in this context?). Does this information solve the question? Is 1111 out of 2016 significant? Is the analysis of the 631 or 1529 promoters conserved across species the best solution? Would not be better a pair-wise comparison between the five species? (All what does not detract from the interesting findings regarding the conservation of the properties of the set of 631 conserved promoters of lncRNAs.)

The highly stage-specific expression patterns of lncRNA promoters are simply described here (Figure 6), they are not interpreted in any way so as to imply biological functionality. Conclusions about functionality are exclusively drawn from analyses of evolutionary conservation at the levels of sequence (Figure 6) and expression specificity (Figure 6).

The procedure to assess TSC functional conservation is described in detail in the Materials and methods section, “TSC conservation” subsection.

The proportion of lncRNAs that is functionally conserved does not, in and of itself, resolve any question – it is just a descriptive statistic.

Asking whether 1,111 conserved TSCs out of 2,016 is significant would in some way test the hypothesis that *all* lncRNA TSCs in *D. melanogaster* are functional, which is not the one we are interested in here.

Instead, we are asking whether those lncRNA promoters that are functionally conserved across all species are functional. We are formally testing this hypothesis with analyses of evolutionary conservation at the levels of promoter sequence (Figure 6) and expression specificity (Figure 6). Both show clear hallmarks of negative selection, implying functionality.

The decision to focus on TSCs conserved in all 5 species is largely technical: it gives us maximal power to accurately estimate expression profile divergence (by comparing expression profiles for more species) and sequence divergence (by computing sequence conservation scores from more genomes).

24) What is less convincing is the final argument based on the analysis of the lncRNA TSCs that are specific to the melanogaster subgroup based the results shown in Figure 6, that shows some similarity between with the conservation profiles of protein coding gene promoters. It seems to be a little be exaggerated, based on the data presented, to conclude "Taken together, our observations show that a vast proportion of lncRNAs are indeed under purifying selection for biological functions relevant to embryonic development."

The observations described in Figure 6 show that, as a class, the lncRNA TSCs that are conserved in all 5 species are under negative selection (just as much as the TSCs of conserved protein-coding genes).

Figure 6 (and Figure 6—figure supplement 5) shows that, as a class, lineage-specific lncRNA TSCs are under lineage specific negative selection (just as much as the TSCs of conserved protein-coding genes). Based on this, we feel it is legitimate to postulate that a large proportion of lncRNA TSCs must be functional.

However, as we indeed cannot provide an exact estimate of the proportion of these that are indeed under selection – all our analyses are inherently class-based – this sentence has been edited as follows:

“Taken together, our observations raise the possibility that a substantial proportion of lncRNAs may indeed be under purifying selection for biological functions relevant to embryonic development.”

25) The arguments on the role of the lncRNA TSCs, based on conservation of a subset of them, are not sufficiently strong to close the debate on the actual role of the lncRNA, even if they may serve to point to the function of some specific cases in development control. Along this line, the argument in the final paragraph of the Discussion section, that the work unambiguously demonstrates the relevance of lncRNAs to development is also a bit strong. The work is highly suggestive, but careful knockouts or other perturbation experiments would be needed to "unambiguously demonstrate" – and even then proving that it isn't actually some short ORF would be tough. I recommend softening this language a bit to avoid raising the hackles of the lncRNA and/or the smORF communities.

To address the reviewers’ concerns, this sentence has been edited as follows:

“Our work provides evolutionary evidence for a broad biological relevance of noncoding transcription to developmental processes, and establishes *D. melanogaster* as a powerful model for exploring the diverse functions of lncRNAs with targeted genetic and molecular studies.”

26) In the third paragraph of subsection “schnurri-like RNA: A deeply conserved, developmentally regulated lncRNA gene”; the claim that the expression pattern is similar to shn seems odd. The gene shn is expressed throughout much of development (certainly through stage 16), it is not restricted to a 3 hour period as with this lncRNA. Here are the in situs of shn: http://insitu.fruitfly.org/cgi-bin/ex/report.pl?ftype=1&ftext=FBgn0003396. Do the authors mean just at this developmental period? If so, a comparison to many other expression profiles, e.g. from the BDGP, would be needed to support the association between this lncRNA and shn – is shn really the "most" similar at this period?

The analogy with *shn* was indeed based on the similarity of expression profiles at the late blastoderm stage only: http://insitu.fruitfly.org/insitu_image_storage/img_dir_117/insitu117893.jpe.

We make no claims that this is the most similar expression pattern observed – it just happens to be similar to that of a well-known gene – and the main text has been modified to move all speculative references to shnurri to the Discussion section.

27) Further, the statements in the final paragraph of subsection “schnurri-like RNA: A deeply conserved, developmentally regulated lncRNA gene” that the authors have "confirmed it's noncoding nature" is too bold. The data is consistent with a non-coding transcript, but it is far from "confirmed". Absence of evidence is not necessarily evidence of absence. And on the contrary, doesn't it look quite cytoplasmic in the FISH? It encodes several ORFs > 20aa – I recommend softening this statement a bit. It looks non-coding, it's probably non-coding, but it could also encode a short ORF – to go the extra mile on this would require targeted quantitative proteomics. Not necessary for publication of course, but without it, the statement should be softened.

This is indeed a thorny issue, and we agree with the reviewers’ concerns. We cannot exclude that this locus does encode a short peptide, or even that this peptide may itself serve a biological function in *D. melanogaster*. However, based on our analyses – and particularly Figure 6—figure supplement 6 – we are making the case that a smORF-associated function cannot account for the conservation of the promoter, as the smORFs themselves are clearly not conserved. This implies the existence of a smORF-independent function that is itself the target of purifying selection. The cytoplasmic localization of the lncRNA in no way detracts from the possibility of a non-coding (possibly cytoplasmic) function.

However, we agree that the language should be modified to more accurately reflect that this constitutes inference, not formal proof. The sentence has been edited as follows:

“Although further characterization will be required to decipher the precise biological role of this transcript, our observations definitively establish its developmental expression specificity, clearly point to a non-protein-coding function, and provide strong evidence that it has been under strong purifying selection over at least 25-50 million years.”

28) The TGF-β link is also quite a leap – it would be good to strengthen this portion of the paper with a bioinformatics analysis of the in situ imaging data against the BDGP database.

We are aware that the paper would be measurably strengthened with the analyses suggested. However, these analyses would require expertise we do not currently possess. Thus, collaborations would be needed. A fuller functional study of this lncRNA is planned but is outside the scope of this paper due to the departure from my laboratory of the graduate student whose thesis work is the subject of this manuscript. Therefore, we are restricting suggestions of a TGF-β pathway link strictly to the Discussion section, merely as speculation about the possible function of the transcript.

29) The observation that specific core promoter elements are associated with TSCs that are activated at specific points during development was made using the subset of TSCs that are functionally active in all species. If the analysis is performed on all TSCs observed in Drosophila melanogaster, are the same trends observed? Or is this observation specific to conserved TSCs?

The question raised by the reviewers is reasonable but the analysis required is not straightforward, particularly regarding the initial classification of promoters – this would require substantial new coding and testing of the results to determine the confidence level of the outcomes, and whether they are comparable to the ones presented here. Furthermore, this classification is chiefly interesting as the basis for the evolutionary analyses that follow, which show that this code is conserved (Figure 4).

We propose to raise this topic in the Discussion as an interesting point to be pursued in subsequent studies.

30) Are the same core promoter element/TFBS trends seen for the non-coding TSCs? In other words, do you see a relationship between the promoter sequence elements for non-coding TSCs and their timing of activation?

Unfortunately, our attempts to conduct this type of analysis were stymied by the fact that there are too few highly-expressed lncRNA TSCs (for which expression divergence can be accurately measured) for us to have any power for the promoter motif analysis. This issue will be resolved in the future with deeper transcriptome data.